# Phase separation of competing memories along the human hippocampal theta rhythm

Casper Kerrén[1,2]*, Sander van Bree[3], Benjamin J Griffiths[1], Maria Wimber[1,3]*

[1]Centre for Human Brain Health, School of Psychology, University of Birmingham, Birmingham, United Kingdom; [2]Research Group Adaptive Memory and Decision Making, Max Planck Institute for Human Development, Berlin, Germany; [3]Centre for Cognitive Neuroimaging, School of Neuroscience and Psychology, University of Glasgow, Glasgow, United Kingdom

**Abstract** Competition between overlapping memories is considered one of the major causes of forgetting, and it is still unknown how the human brain resolves such mnemonic conflict. In the present magnetoencephalography (MEG) study, we empirically tested a computational model that leverages an oscillating inhibition algorithm to minimise overlap between memories. We used a proactive interference task, where a reminder word could be associated with either a single image (non-competitive condition) or two competing images, and participants were asked to always recall the most recently learned word–image association. Time-resolved pattern classifiers were trained to detect the reactivated content of target and competitor memories from MEG sensor patterns, and the timing of these neural reactivations was analysed relative to the phase of the dominant hippocampal 3 Hz theta oscillation. In line with our pre-registered hypotheses, target and competitor reactivations locked to different phases of the hippocampal theta rhythm after several repeated recalls. Participants who behaviourally experienced lower levels of interference also showed larger phase separation between the two overlapping memories. The findings provide evidence that the temporal segregation of memories, orchestrated by slow oscillations, plays a functional role in resolving mnemonic competition by separating and prioritising relevant memories under conditions of high interference.

*For correspondence:
casper.kerren@gmail.com (CK);
Maria.Wimber@glasgow.ac.uk
(MW)

**Competing interest:** The authors declare that no competing interests exist.

## Editor's evaluation

This pre-registration study used a proactive interference task in combination with MEG recordings on humans to test predictions of a previous computational model postulating that neural representations of competing memories are associated with varied phases of hippocampus theta-band rhythm. Their results confirmed the hypothesis and suggest that reactivations of target and competitor memories indeed occur at different phases of theta oscillations, which is further related to the intrusion effect in behavior.

## Introduction

Each day is a flow of events that take place at different times but often in overlapping contexts. Unavoidably, many of the stored memories share similar features, and this overlap poses a major challenge for our memory system (*McClelland et al., 1995*; *Norman and O'Reilly, 2003*). The present magnetoencephalography (MEG) study investigates the possibility that the human brain uses a

temporal phase code to adaptively separate overlapping memories, enabling the targeted retrieval of goal-relevant information.

Competition between similar memories is considered one of the major causes of forgetting (*Anderson and Neely, 1996*; *Underwood, 1957*). A prominent case is proactive interference, where access to a target memory is impaired when overlapping information has been stored prior to target learning (*Tulving and Watkins, 1974*). This impairment is typically ascribed to the conflict arising from the co-activation of competing memories (*Kliegl and Bäuml, 2021*). On a neurophysiological level, mid-frontal theta oscillations (3–8 Hz) have been identified as a reliable marker of cognitive conflict in general (*Cavanagh and Frank, 2014*) and mnemonic conflict specifically (*Ferreira et al., 2014*; *Hanslmayr et al., 2010*; *Johansson et al., 2007*; *Staudigl et al., 2010*). Increased theta amplitudes may facilitate the emergence of phase separation, where populations of neurons transmitting potentially interfering information are segregated in time along a theta cycle (*Buzsáki and Draguhn, 2004*).

On a theoretical level, computational models assign a central role to the phase of the hippocampal theta oscillation in the ordered representation of multiplexed information (*Lakatos et al., 2005*; *Lisman, 2005*; *Lisman and Idiart, 1995*; *Lisman and Jensen, 2013*; *Nyhus and Curran, 2010*; *O'Keefe and Recce, 1993*). Empirically, in rodents it has been shown that bursts of gamma power reflecting sequences of spatial information processing are represented at different phases of the hippocampal theta oscillation (*Bragin et al., 1995*; *Colgin et al., 2009*). Several recent studies in humans suggest that the neural representations of items held in working memory or encoded into or retrieved from episodic memory (*Kerrén et al., 2018*; *Kunz et al., 2019*; *Pacheco Estefan et al., 2021*) cluster at distinct phases of the (hippocampal) theta rhythm. These findings suggest that low-frequency oscillations provide time windows for the selective processing and readout of distinct units of information (*Lisman and Jensen, 2013*). This study investigates phase separation as a potential mechanism to avoid interference between multiple competing memories that are simultaneously reactivated by a cue.

The computations by which the human brain achieves a separation of multiple overlapping memories are currently unknown. One computational model leverages different phases of a theta oscillation to iteratively differentiate target from competitor memories (*Norman et al., 2006*). In this model, which we will henceforth refer to as the *oscillating interference resolution model*, a retrieval cue will activate associated units, representing target and competitor features, in a phase-dependent manner. In the most desirable output state, at medium levels of inhibition, the cue only activates the target units and no or few competitor features. When inhibition is raised towards the peak of the oscillation, only strong features of the target memory will remain active, and the model learns (via contrastive Hebbian learning) to strengthen through long-term potentiation (LTP) the weaker target nodes that did not survive the higher inhibition levels. Conversely, during the transition from a medium to a lower inhibition state towards the trough of the oscillation, activation spreads to more weakly associated units, including some competitor features. This opposite phase is used to identify and punish overly strong features of the competing memory through long-term depression (LTD). The mechanism (see *Figure 1*) is repeated across several cycles of an oscillation, which changes the similarity structure of memories into a state in which they are less likely to interfere with each other. This model offers a mechanistic but not yet tested explanation for how the brain solves mnemonic competition.

A previous study (*Kerrén et al., 2018*) provided proof-of-principle evidence that the reactivation of a single visual memory is rhythmically modulated by the hippocampal theta rhythm. In the present study, we applied the same analytic approach in an AB-AC associative interference paradigm (*Figure 1a and b*) to investigate the phase modulation of competing memories. Hypotheses for the experiment were pre-registered on OSF. Based on the oscillating interference resolution model (*Norman et al., 2006*) and related theta-gamma models (e.g., *Lisman and Idiart, 1995*), we expected that target and competitor memories become active, and can thus be decoded, at different phases of the hippocampal theta oscillation, with only target features active at higher inhibition phases, and competitors surfacing as inhibition ramps down to medium and low levels (see *Figure 1d*). The model also assumes that across several repetitions of the theta cycle, weak features of target memories are strengthened while overly strong features of competing memories are punished to make them less interfering on future cycles. We hypothesised that the effects of these dynamics will become visible across several repetitions of competitive retrieval and lead to an increasing phase separation because over time target memories become more likely to activate in the high-inhibition phase whereas

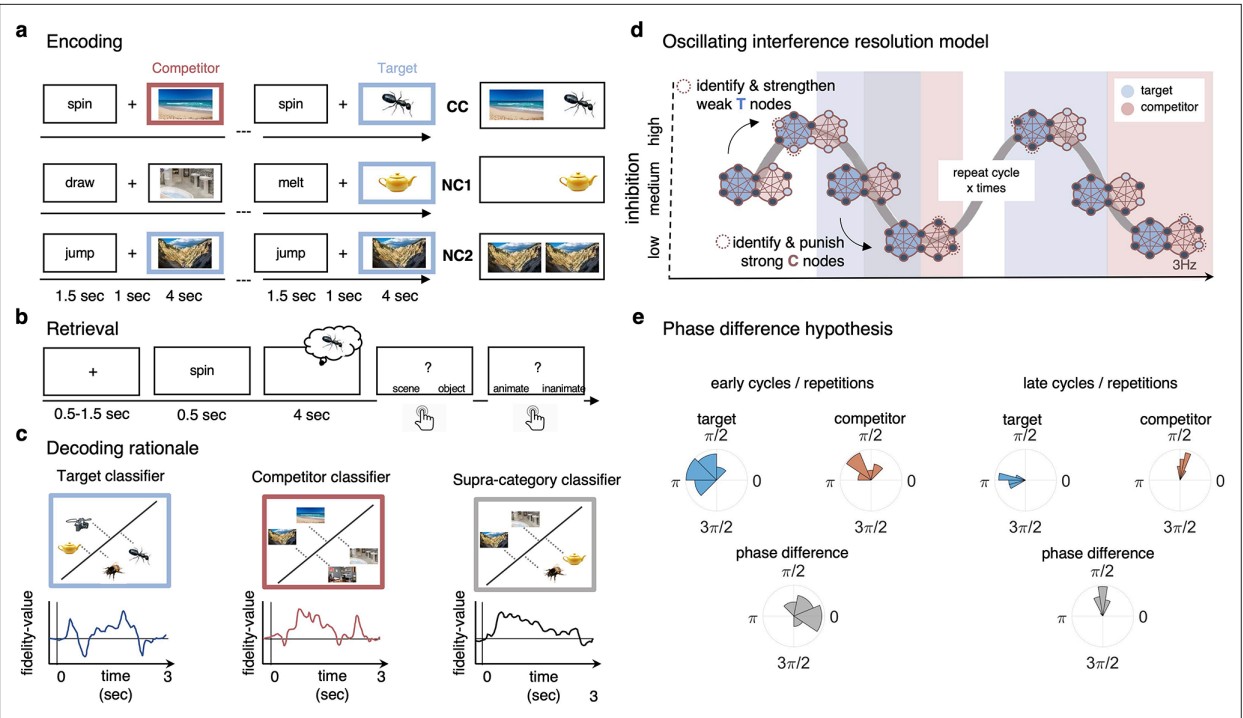

**Figure 1.** Paradigm and rationale for decoding analyses. (**a**) At encoding, subjects (n = 24) were instructed to memorise the word–image associations using an imagery strategy and constantly update their memory with the most recent associate to each word. The experiment consisted of three different conditions. In non-competitive single-exposure condition (NC1), subjects encoded the word together with one associate; in the non-competitive dual-exposure condition (NC2), subjects encoded the word with the same associate twice; and in the competitive condition (CC), subjects encoded the word together with two different associates (one scene and one object). (**b**) At retrieval, participants were instructed to remember the most recently encoded associate when prompted with a word cue. (**c**) Subordinate-category classifiers (animate/inanimate for objects, and indoor/outdoor for scenes) were used to obtain independent evidence for target and competitor reactivation at each sample point. Note that the superordinate (object/scene) classifier cannot discriminate between evidence for the target and against the competitor, and vice versa. (**d**) Hypothesised reactivation dynamics of targets and competitors relative to theta phase, based on the oscillating interference resolution model (adjusted from **Norman et al., 2006**). Blue circles represent the target memory, pink circles the competitor memory. Target and competitor memories consist of a number of features (small circles), some distinct and some overlapping, that can be either active (dark blue) or inactive (light blue) at any given point along the phase of the oscillation. When inhibition is high, only a few features of the target memory are active initially. The transition from medium to high levels of inhibition is used in the model to identify and strengthen weak target features, such that these nodes have a higher likelihood of becoming active in future high-inhibition phases (see right portion of the graph). In contrast, during the low-inhibition phase, features of both target and competitor memories are simultaneously active. The transition from medium to low levels of inhibition is used to identify and punish overly strong competitor features, which will in turn be less likely to activate even in low-inhibition phases in the future. (**e**) An illustration of the expected phase of maximum target and competitor reactivation, and their phase difference, in the first and last repetition of retrieval. Note that in this example target and competitor memories show a consistent phase angle across subjects in the third repetition; however, this is not a necessary assumption for finding consistent phase separation.

competing memories require increasingly lower levels of inhibition to activate. Lastly, in line with previous literature (**Wimber et al., 2015**), the strengthening of target and weakening of competing memories should also be reflected in corresponding changes in their decodability.

## Results

Participants (n = 24) completed an associative memory task including one proactive interference condition and two control conditions (**Figure 1a and b**). In each learning trial, participants were instructed to form a novel association between a unique verb and an image, which could be either an object or a scene. In the critical competitive condition (CC), the word was associated with two images, one object and one scene image, learned on different trials. Participants were asked to continuously update their memory to remember the most recently encoded associate of each word, thus allowing us to investigate the extent to which previously encoded associations interfered with newly encoded information (proactive interference). In the single-exposure non-competitive condition (NC1), a word

was associated with only one image, and never reappeared during learning. This condition served as a baseline for behavioural interference effects. In the dual-exposure non-competitive condition (NC2), a word was also associated with one image only, but this association was learned twice. This condition was included as a control for neural (cue) repetition and order effects. Both non-competitive conditions were also used to train and validate our multivariate classifiers. CC trials were pseudo-randomly intermixed with NC1 and NC2 trials during learning and recall blocks.

In a subsequent memory test, participants were asked to remember the most recent image when prompted with the cue word, and cued recall was repeated three times per word–image association, in a spaced fashion, irrespective of condition. Once participants indicated, via button press, that they had the associated picture back in mind, two follow-up questions were asked, one about the supraordinate (object/scene) and one about the subordinate (animate/inanimate for objects, indoor/outdoor for scenes) category. This task design, with its supra- and subordinate categories, is critical because it allowed us to disentangle, on a neural level, the representational evidence for target and competitor reactivation at any given time point, using two independently trained linear decoding algorithms: one to discriminate indoor and outdoor scenes, and one to discriminate animate and inanimate objects (see example in *Figure 1c*).

## Behavioural indices of proactive interference

We first evaluated behavioural evidence for proactive interference, that is, the extent to which cued recall performance suffered from having encoded two different pictures with one word (CC) compared to only one picture (NC1). Only trials where participants correctly responded to both follow-up questions (i.e. they were able to provide the supra- and subordinate category of the target image) were considered a successful recall. A one-factorial repeated-measures ANOVA, with memory accuracy as the dependent variable, and condition (CC, NC1, NC2) as within-subject factor, showed a significant main effect of condition, $F(1.453, 46) = 57.99$, $p<0.05$. The main planned comparison of interest showed that memory performance was indeed significantly reduced in the competitive (M = 53.18%, SD = 19.20%) compared to the single-exposure (M = 57.71%, SD = 13.98%) condition, $t(23) = 1.80$, $p<0.05$ (*Figure 2a*). This confirms the first behavioural hypothesis (OSF) that retrieving a memory is more difficult when a previously encoded, now irrelevant associate competes for recall, indicative of proactive interference. For later analyses relating neural and behavioural indices, an interference score for each participant was calculated by subtracting average memory performance in the CC from the NC1 condition. A second post-hoc test showed that a significant difference was also obtained between NC1 and NC2 (M = 78.88%, SD = 10.06%), $t(23) = -11.91$, $p<0.05$, demonstrating that, unsurprisingly, a second encoding exposure improved performance compared to a single exposure.

Another behavioural index of competition is the intrusion score; a participant's tendency to choose the competitor instead of the target. Our design allowed us to quantify intrusions based on how often participants selected the specific subcategory of the competitor (i.e. 'outdoor scene' in the example in *Figure 1*) during the staged retrieval. On trials where participants made an error (i.e. selected 'object' though the target was a scene and vice versa), they showed a significantly above-chance (i.e. above 50%) probability of selecting the subcategory of the competitor (on average, 88.72%, SD = 10.62%; $t(23) = 10.48$, $p<0.05$), suggesting that their errors were not random but biased towards the category of the previously learned but now irrelevant associate (*Figure 2b*). The proportion of intrusions did not change significantly across repetitions (first: 88.08%, SD = 12.04%; second: 89.23%, SD = 10.62%; third: 88.85%, SD = 14.72%, $Z = -54$, $p=0.59$; Wilcoxon signed-rank test of linear slope against zero). Similarly, we did not find a decrease across repetitions in behavioural accuracy for CC target items ($Z = -0.73$, $p=0.47$; Wilcoxon signed-rank test of linear slope against zero; see *Figure 2—figure supplement 1a*). Across participants, the average number of intrusions in the CC correlated negatively with memory performance in the non-competitive conditions, $r = -0.37$, $p=0.037$, $p<0.05$ (see *Figure 2—figure supplement 1b*), indicating that better memory performance was generally associated with fewer intrusion errors.

The average response time for subjective recollection (indicated by button press) was 2.11 s (SD = 0.15 s) when collapsing across all conditions and repetitions. Response times shortened across repetitions in all three conditions (CC: rep1: 2.29 s, rep2: 2.05 s, rep3: 2.04 s; $Z = -2.89$; $p=0.004$; NC1: rep1: 2.55 s, rep2: 2.37 s, rep3: 2.26 s; $Z = 3.03$; $p=0.003$; NC2: rep1: 2.01 s, rep2: 1.75 s, rep3: 1.68 s; $Z = -4.17$, $p<0.001$; Wilcoxon signed-rank tests of linear slope against zero), suggesting that the

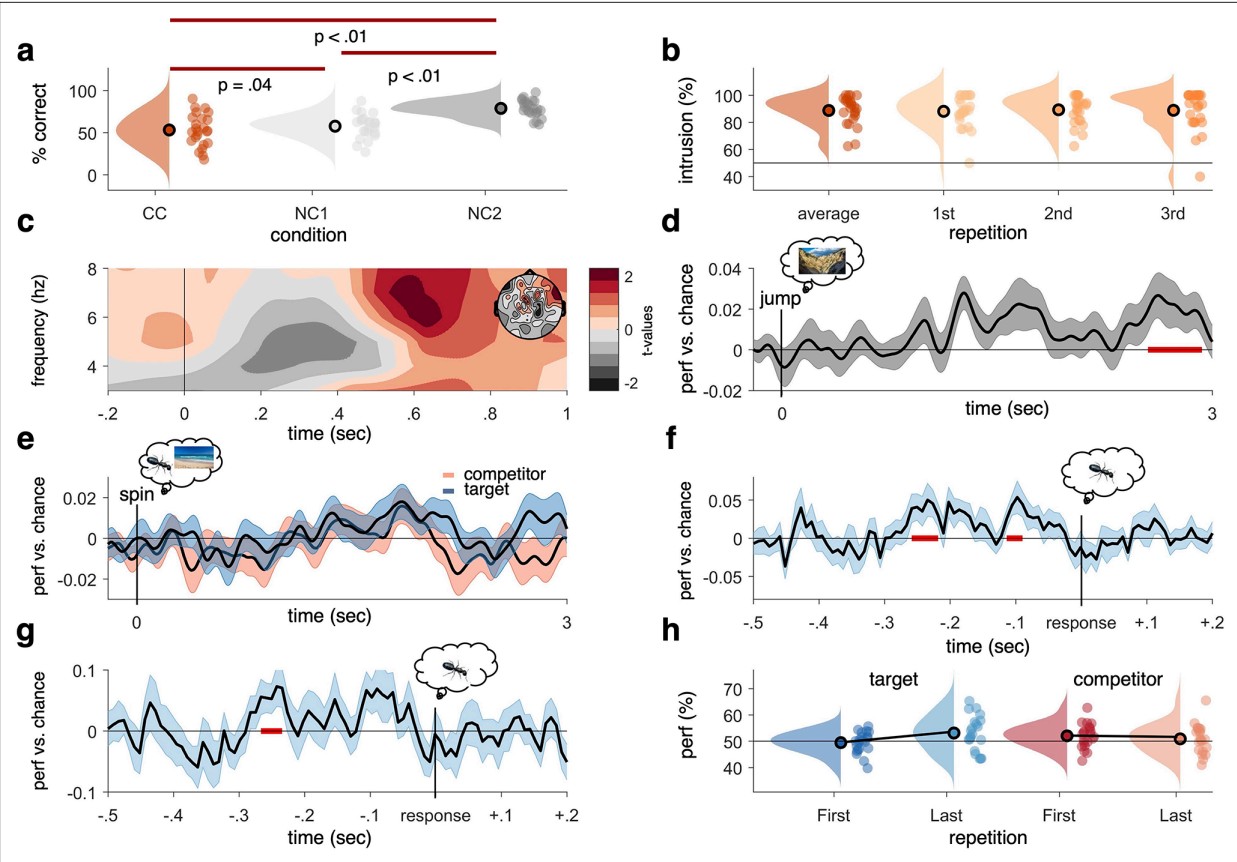

**Figure 2.** Behavioural results, time–frequency analysis of theta power, and decoding accuracies in the non-competitive (NC) and competitive conditions (CCs). (**a**) As expected, we found that memory accuracy (on the two follow-up questions combined), averaged across the three recall repetitions, was significantly impaired when encoding a given word cue with two different images (CC) compared to just one image (NC1), indicative of proactive interference (red line indicating significant difference at p<0.05, using Paired Samples t-test). Recall performance also benefited from learning a cue word together with the same image twice (NC2) compared to once (NC1). (**b**) The average intrusion score shows that errors were not random (50% black line), but instead were significantly biased towards the competitor's subcategory. The proportion of intrusions did not decrease significantly across repetitions. (**c**) Contrasting oscillatory power elicited by the cue in the CC and NC2 conditions resulted in a significant cluster (500–700 ms, darkest red, cluster-based permutation < .05) in the theta frequency range (3–8 Hz), most prominently over right frontal electrodes (see right inlay). (**d**) Results of a linear discriminant analysis (LDA)-based classifier trained and tested on the non-competitive conditions (NC1 and NC2) at retrieval, showing a cluster of significantly above-chance decoding accuracy approximately 2500–3000 ms post-cue onset, with an earlier decoding peak around 1–2 s not surviving cluster correction. Graph shows mean and SEM. (**e**) Results from a classifier trained on the non-competitive conditions (NC1 and NC2) and tested on the CC. Separate classifiers were used to detect target (blue) and competitor (red) evidence at the level of subcategories. No significant cluster emerged when averaging across all repetitions. (**f**) Realigning the trials to the time of subjective recollection (i.e. response), instead of cue onset, significant target decoding was found in the CC when averaging over all repetitions. (**g**) Response-locked target decoding in the CC was also significantly higher on correct than incorrect trials. (**h**) Using a repeated measures ANOVA to con decodability in the first and third retrieval repetition for target and competitor memories, respectively, yielded a significant interaction such that evidence for target memories increased as a function of repetition, whereas evidence for competitor memories decreased as a function of repetition. For behavioural analyses, 24 participants were included, whereas for neurophysiological analyses 21 participants were included. Red line in panel (**d, f, g**), indicates time windows of significant clusters at p_cluster<0.05 corrected for multiple comparisons across time.

The online version of this article includes the following figure supplement(s) for figure 2:

**Figure supplement 1.** Behavioural performance across repetitions, correlation between intrusion score and memory performance, and correlation between intrusion score and decoding performance.

**Figure supplement 2.** Additional decoding analyses during encoding and retrieval.

time point when participants remembered the associate shifted and overall was faster after repeated recalls, as would be expected.

## Mnemonic competition is associated with a frontal theta power increase

Having established that the paradigm elicits competition on a behavioural level, we next checked whether the CC was associated with the neural indices of competition typically reported in the literature. A well-documented effect is an early increase in theta power (3–8 Hz) over frontal sensors, which has been related to cognitive conflict processing in tasks with and without memory demands (*Cavanagh and Frank, 2014*; *Ferreira et al., 2014*; *Hanslmayr et al., 2010*). In this study, the CC and the non-competitive neural baseline (NC2) were equated for repetitions of the cue. When contrasting these two conditions, we expected to see a similar power increase in the theta range in the present experiment. A non-parametric cluster-based permutation test, focused on a time window of interest 0–1 s following cue onset and on the previously reported 3–8 Hz theta range, revealed a significant power increase when retrieving the target association in the CC compared to the NC2 condition ($p_{cluster}<0.05$), ranging from approximately 500 to 700 ms post-cue (*Figure 2c*). In addition to behavioural evidence for proactive interference, we thus also replicate previous work that established frontal theta power increases as a neural signature of competition. Note, however, that in terms of its topography, this frontal effect was more right-lateralised than the typically observed mid-frontal theta (*Cavanagh and Frank, 2014*), and might instead be more related to response conflict or response inhibition (for a review, see *Aron et al., 2004*). The relationship of this faster frontal to the typically slower hippocampal rhythm (see below) is currently unclear, and it would be an interesting target for future studies to investigate whether the two rhythms relate to a common pacemaker (e.g. the medial septum, see *Wang, 2002*).

## Pattern reactivation of target and competitor memories

The main pre-registered hypothesis of this study concerned the timing of neural reactivation of target and competitor memories relative to the phase of the theta rhythm. As a first step, we thus checked whether and when the categorical content of the associated memories could be decoded from the MEG sensor patterns in the recall phase of the experiment. Two separate linear discriminant analysis (LDA)-based classifiers were trained to discriminate the subordinate picture class, that is, animate vs. inanimate for objects, and indoor vs. outdoor for scenes. As noted above, the two subordinate categories were an important feature of the task design that allowed us to obtain independent indices of target and competitor reactivation. Both classifiers were trained on distinguishing subcategory membership (animate/inanimate and indoor/outdoor for objects and scenes, respectively) in the noncompetitive ('pure') conditions during recall.

As a sanity check, we then first tested these classifiers' ability to detect the retrieval of objects and scenes in the non-CCs (NC1 and NC2). Training and testing were conducted time point per time point, in steps of 8 ms, each time point centred in the middle of a Gaussian smoothing window (fullwidth at half maximum [FWHM] = 40 ms). MEG gradiometer sensor patterns (i.e. signal amplitude on each of the 204 channels) were used as classifier features, independently per participant and per time bin (from –500 to 3000 ms around cue onset). Since the same data was used for training and testing in this case, a tenfold cross-validation was used and repeated five times. Results are shown in *Figure 2d*, where highlighted in red are significant time points corrected for multiple comparisons over time using a non-parametric cluster-based permutation test (as implemented in FieldTrip toolbox version 2019, *Oostenveld et al., 2011*). This first decoding analysis indicated that reliable ($p_{cluster}<0.05$) above-chance decoding performance was found in a time window from 2.55 to 2.92 s post-cue onset, somewhat later than the typical time window for reinstatement of associative memories in cued recall tests (for a review, see *Staresina and Wimber, 2019*). Earlier clusters did not survive stringent cluster correction. For decoding during visual presentation of the stimuli at encoding, see *Figure 2—figure supplement 2a*.

To test for reactivation of target and competitor memories in the CC, trials were split into those where the target was an object and the competitor was a scene, and vice versa. The corresponding object (animate vs. inanimate) and scene (indoor vs. outdoor) classifiers, again trained on noncompetitive recall trials, were then used to indicate evidence for target and competitor reactivation,

restricted to only correct trials. *Figure 2e* shows the average target decoding accuracy in the CC. Classification performance peaked in time windows similar to the non-competitive condition. However, no significant cluster could be identified when correcting for multiple comparisons across time points, indicating that overall reinstatement was less robust in the condition where two associates competed for recall, potentially because this competition was associated with less confident target retrieval. A more technical explanation for the absence of significant target decoding in the CC is the variance in the timing of subjective recollection across trials and participants, making it difficult for a classifier to detect a consistent time point of neural reinstatement. We thus conducted an additional analysis (not pre-registered, added as part of peer review process) realigning the decoding data to the subjective recollection button press, and thus to the time point when participants indicated they had the associated image back in mind. This method revealed two clusters of significant target classification preceding the button press by approximately 200 ms (*Figure 2f*). The realigned analysis further revealed that target classification on correct trials was significantly stronger than on incorrect trials (*Figure 2g*). These results reaffirm that the classifiers are able to pick up categorical reactivation also in the CC, and that this reactivation varies with behavioural performance in a meaningful way. They also suggest that the lack of significant decoding in the cue-locked analyses is due to timing differences in memory reinstatement between conditions, trials, and participants, which is rectified when locking to the time point of subjective recollection.

In addition to general decodability when averaging across all retrieval repetitions, we also predicted in our pre-registration that in the CC the neural representation of the target memories would become stronger and the competing associate would become weaker across repetitions (see *Wimber et al., 2015*). A contrast of the decoding timelines between the first and third repetition revealed a significant increase ($p_{cluster}$<0.05, 1.79–2.11 s; third repetition compared to chance $p_{cluster}$=0.07) in decodability of the target memory approximately 2 s after cue onset (*Figure 2—figure supplement 2b*). No corresponding cluster emerged when comparing the entire timeline of competitor reactivation between the first and third recall (*Figure 2—figure supplement 2c*). An additional analysis testing for competitor evidence on incorrect trials, when competitors are more likely to interfere, revealed more robust competitor activation (see *Figure 2—figure supplement 2d*). Zooming in on the time window where targets and competitors were maximally decodable (1.77–1.93 s after cue onset, $p_{uncorr}$<0.05, see *Figure 2—figure supplement 2e*), a significant repetition-by-associate interaction was found ($F$(1,20) = 5.187, p=0.034), with a significant increase of target reactivation from first to third recall (first: 49.43%, SD = 4.08%; third: 53.14%, SD = 6.43%; $t$(20) = –2.46, p=0.02), and a numerical but non-significant decrease of competitor reactivation (first: 52.05%, SD = 4.55%; third: 50.82%, SD = 5.54%; $t$(20) = 1.1, p=0.29) (*Figure 2h*). Thus, we observed a strengthening of the relevant target memory and a numerical but non-significant weakening of the competing memory across repetitions in this select time window. Lastly, taking inspiration from previous work on competitor reactivation and its relationship to behavioural memory interference (*Newman and Norman, 2010*), we extracted the average decoding across repetitions for the competitor memories from 0 to 1 s after cue onset and correlated this reactivation with the probability of intrusions in the first repetition. This revealed a significant correlation ($r$ = 0.446, p=0.04; *Figure 2—figure supplement 1c*), such that stronger reactivation of competitor memories positively correlated with a higher level of initial intrusions.

Together, the cue-locked decoding results provide evidence that target memories can be decoded from MEG signals when no competition is present, while under conditions of interference, target reactivation is less robust in early trials but gradually increases over repetitions. This pattern could emerge due to the gradual strengthening of the relevant target memory or due to reduced interference from the competing memory. When instead realigning the decoding to the subjective time of recollection, we found significant target decoding, indicating that temporal variance in memory reactivation could explain the non-significant results in the cue-locked analysis.

## Hippocampal theta oscillation clocks the reactivation of target and competitor memories

The next step was to assess whether the fidelity with which target and competitor memories can be decoded fluctuates in a theta rhythm. The analysis was conducted to determine the dominant frequency in the decoding timelines, to replicate our previous findings for target memories (*Kerrén et al., 2018*), and to extend the principle to competitor memories. We estimated the

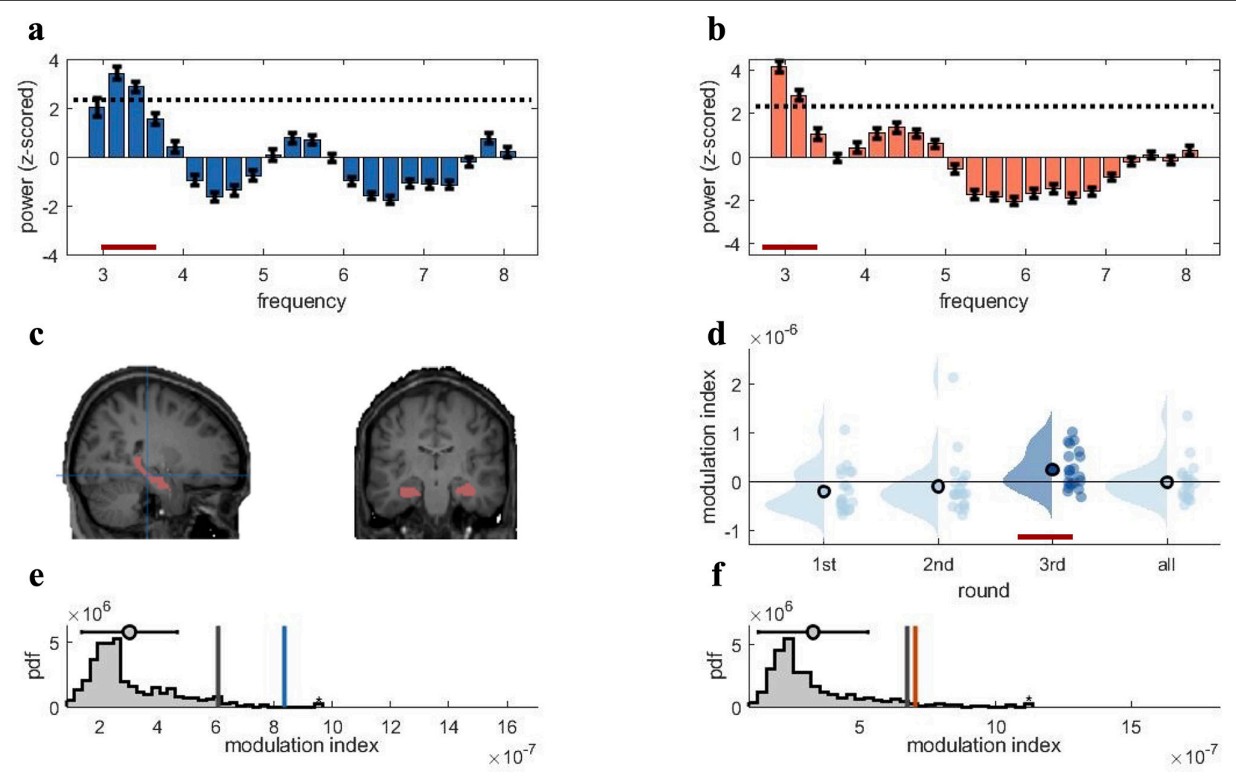

**Figure 3.** Rhythmic fluctuations in classifier fidelity and phase locking to hippocampal theta oscillation. (**a, b**) An estimation of oscillatory components in the classifier fidelity values, using Irregular-Resampling Auto-Spectral Analysis (IRASA), revealed significant rhythmicity at 3 Hz, compared with the same estimate from a label-shuffled baseline classifier (dashed black line; red line indicates frequencies that significantly differ from baseline) for both target (**a**) and competitor (**b**) memories. (**c**) The hippocampal region of interest used as the source for theta phase, shown in sagittal and (left) coronal (right) plane. (**d**) Distribution of each individual's modulation index (MI) with the 95th percentile subtracted. The third retrieval repetition shows significant modulation exceeding the 95th percentile (red line). (**e, f**) Results of the phase modulation analysis linking decoding fidelity to the phase of the hippocampal 3 Hz oscillation. Fidelity values for target (**e**) and competitor (**f**) memories were significantly modulated by the hippocampal theta rhythm at 3 Hz. The histogram shows the chance distribution of the MI, together with the 95th percentile in grey vertical bars and empirical value in blue for target memories and orange for competitor memories. For all analyses, n = 21 participants were included. Graphs in a and b show means and SEM (error bars).

The online version of this article includes the following figure supplement(s) for figure 3:

**Figure supplement 1.** Frequency profile of fidelity timecourses across repetitions.

**Figure supplement 2.** Control analyses preprocessing and source localisation.

oscillatory component of the fidelity values, following our pre-registration and the rationale of our previous study (***Kerrén et al., 2018***). We used Irregular-Resampling Auto-Spectral Analysis (IRASA) (as implemented in the FieldTrip toolbox version 2021), a method that robustly detects and separates the oscillatory from the fractal signal component in electrophysiological data (***Wen and Liu, 2016***). IRASA was applied to the single-trial fidelity timecourses in the CC, in a time window from 500 to 3000 ms post-cue onset, and separately for the target and competitor memories. The first 500 ms were excluded to attenuate the influence of early, cue-elicited event-related potentials, and because no neural reinstatement is expected during this early cuing period (***Staresina and Wimber, 2019***). The analysis identified 3 Hz as the frequency with the highest power for both target and competitor memories (same peak frequency was also observed for the first repetition; ***Figure 3—figure supplement 1a and b***), and power at 3 Hz exceeded the 95th percentile of the empirical chance distribution generated from surrogate-label classifiers (***Figure 3a and b***, red line indicating significant deviation). For the NC, the peak frequency was also in the low theta range but slightly faster (4–5 Hz; see ***Figure 3—figure supplement 1c and d***). Since our theta-locked analyses were focused on the CC, we used 3 Hz as the modulating frequency in all subsequent analyses.

## Target and competitor reactivations peak at distinct theta phases

The key hypothesis was that over time the reactivated representations of target and competitor memories would become optimally separated along the phase of the theta oscillation. This hypothesis follows from the oscillating interference resolution model motivating our work (*Norman et al., 2006*) and other empirical studies on retrieval-induced forgetting (RIF; e.g. *Anderson et al., 1994*; *Wimber et al., 2015*). The model assumes that target and competitor representations are overlapping initially, such that strong competitor nodes can (incorrectly) activate during the high-inhibition ('target') phase of theta, but gradually get weakened and thus require lower levels of inhibition to become active. Meanwhile, the weakest target nodes do not survive high inhibition initially and thus only activate during a lower inhibition phase of the theta cycle; however, with repeated strengthening they will become active at an increasingly early, higher-inhibition phase. Therefore, while early in time the target and competitor features overlap in their reactivation phase, this overlap is reduced by the strengthening and weakening dynamics in the model. Note that the original model (*Norman et al., 2006*) predicts a gradual increase of target–competitor separation with each oscillatory cycle of excitation–inhibition. In our pre-registration, we instead hypothesised that the phase separation would be measurable when comparing early with late recall repetitions, allowing us to obtain robust trial-by-trial indices of target and competitor reinstatement and avoid confounds with time-on-trial. As also noted in our pre-registration, there is an influential theta phase model (*Hasselmo et al., 2002*) suggesting that encoding and retrieval operations in the hippocampal circuit are prioritised at opposite phases of the theta rhythm (for evidence in humans, see *Kerrén et al., 2018*). Taking this model into account, we thus hypothesised that the target–competitor phase segregation would over time and repetitions become optimal within the retrieval portion (e.g. half) of the theta cycle, rather than spread out across the entire cycle (not shown in *Figure 1d*).

The phase binning method used to calculate the MI (see above) allowed us to determine the theta phase bin at which target and competitor reinstatement was maximal in a given trial and condition. Importantly, the absolute phase of reinstatement in terms of its angle is likely to vary across participants (e.g. due to individual differences in anatomy). We therefore contrasted the phase of maximal target and competitor reinstatement within each individual participant, computing their phase distance as an index of phase separation. This phase distance is expected to be consistent across participants irrespective of the absolute angle of target and competitor reactivation, and can thus be subjected to group-level statistics. We used a Rayleigh test for non-uniformity (i.e. clustering) to test how coherent the phase separation angle was across participants and a circular v-test to establish whether the mean separation angle significantly deviated from zero. Note that without significant clustering it is difficult to interpret the mean angle in such a phase analysis. On the other hand, significant clustering without a significant difference from zero would indicate that consistently across participants, targets and competitors reactivate at a similar theta phase.

*Figure 4a* shows the results of this phase separation analysis. We found a consistent phase difference between target and competitor reactivation exclusively in the last repetition (Rayleigh test for non-uniformity, $z(20) = 5.03$, p=0.005), clustered around a mean angular difference of 34°. Furthermore, a test against zero phase shift confirmed that this target–competitor distance was significantly different from zero difference ($z(20) = 8.48$, p=0.004). This was not the case for all repetitions averaged, ($z(20) = 0.84$, p=0.44), nor for the first ($z(20) = 2.19$, p=0.11) or second ($z(20) = 1.76$, p=0.17) repetition separately (*Figure 4a*). Furthermore, there was no significant difference in target–competitor phase separation between the first and third repetition when comparing the difference in mean angle ($z(20) = 1.09$, p=0.29). Note, however, that there was no significant clustering around a stable mean angle in the first repetition (see above), and this statistical comparison is therefore inconclusive. Finding the separation between target and competitor memories only in the last repetition rather than throughout the entire recall phase deviates from our pre-registered hypothesis, although paralleling the phase modulation results reported above. Overall, the phase distance analysis thus partly confirms our hypotheses, indicating that competing memories become increasingly separated along the hippocampal theta rhythm over time, with significant phase separation emerging after several recall cycles.

The next set of analyses was not pre-registered and tested for a relationship between target–competitor phase distance and behaviour. The sample was split according to each individual's intrusion score in the third repetition, dividing participants into a low-intrusion and a high-intrusion group (*Figure 4b*). We found that the high-intrusion group had a mean separation angle of 7° that was not

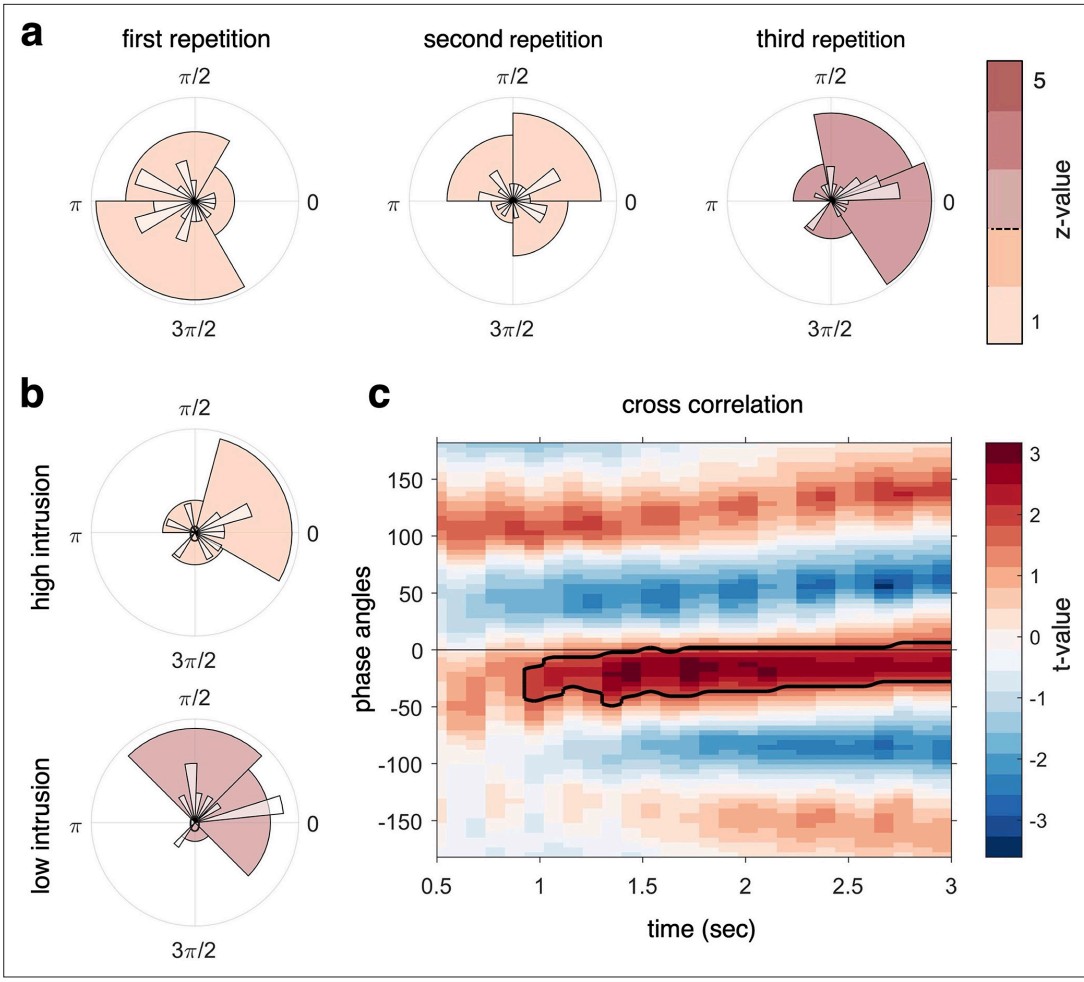

**Figure 4.** Target–competitor phase difference results. (**a**) Phase difference of maximal target minus competitor reactivation for each participant, separately for each recall repetition. At repetition 3 (right circular plot), there was a phase shift of on average 34° that was statistically coherent across subjects and significantly different from zero, indicative of robust target–competitor phase segregation. (**b**) When splitting the phase difference in the third repetition by behavioural intrusion score, the high-intrusion group had a mean phase separation angle close to zero (top), while the low-intrusion group had a mean separation angle of 57° (bottom). (**c**) A cross-correlation between the continuous fidelity timecourses of target and competitor memories revealed a significant cluster ($p_{cluster} < .05$) from approximately 1 s after cue onset, lasting to the end of the trial, with a maximum around 30° phase lag. z-values in panels (**a, b**) (colour code) indicate strength of Rayleigh test for non-uniformity, with z-values/colours exceeding the dashed line on the colour bar indicating significant coherence in the target–competitor phase difference. For all analyses, n = 21 participants were included.

The online version of this article includes the following figure supplement(s) for figure 4:

**Figure supplement 1.** Single subject fidelity values and phase difference.

significantly different from zero (Rayleigh test for non-uniformity, $z(9) = 1.77$, p=0.17). The low-intrusion group, by contrast, showed a mean phase separation of 57° that significantly clustered around this angle (Rayleigh test for non-uniformity, $z(9) = 3.74$, p=0.02); however, we found no significant deviation from zero phase shift in this subgroup of participants (Rayleigh test for non-uniformity with a specified mean angle of 0, $z(9) = 3.32$, p=0.069). A statistical comparison of the phase separation angle in the high- and low-intrusion groups (Wilcoxon signed-rank test in non-circular space, see 'Materials and methods') did not indicate a difference between the mean of the high- and low-intrusion groups' separation angle ($Z = 43$, p=0.85). Note that each subsample in this comparison only contains 10 participants, and the statistical comparison is thus likely underpowered. In sum, this analysis suggests that only low-intrusion participants exhibit significant phase separation by the end of the task, which

can be taken as an indication that more neural differentiation of overlapping memories along the theta cycle relates to more successful behavioural differentiation.

Two more analyses were conducted to corroborate the finding of a phase difference between target and competitor reactivation, complementing the pre-registered analysis reported above. First, we tested for a temporal difference (lag) between the timelines of target and competitor reactivation using a cross-correlation. To do so, we filtered the two fidelity timecourses from each participant at the dominant 3 Hz frequency, in the third repetition of the CC. We then quantified a temporal lag between timecourses by using a sliding window (allowing us to see phase lags that evolve over time) and calculating the cross-correlation for each 330 ms time bin (1/3 of the dominant 3 Hz frequency). Note that this analysis does not specifically take the phase of virtual hippocampal sensors into account and instead simply cross-correlates the entire target and competitor fidelity timecourses derived from all MEG sensors. The analysis revealed a significant cluster starting approximately 1 s after cue onset and lasting until the end of the trial, with a maximum lag of 30° phase angle (*Figure 4c*). This finding suggests that the reactivation of competitor memories is lagging 30° behind the reactivation of target memories.

The second additional analysis used the average target and competitor fidelity timecourse of each participant, filtered at 3 Hz and shown individually in *Figure 4—figure supplement 1a*. Visual inspection already suggests a temporal shift, and sometimes even phase opposition, between target and competitor decoding in most participants. To formally quantify the phase shift on a group level, we subtracted in complex space, for each time bin, the phase angle of the target from the angle of the competitor. The resulting phase separation (*Figure 4—figure supplement 1b*) is consistent with the cross-correlation results shown in *Figure 4c* and suggests a continuous lag of approximately 45° between the two competing memories. The results of these two analyses on the continuous fidelity timecourses are thus consistent with the phase shift identified when looking at the classifier-peak-to-hippocampal-theta locking, and provide further evidence for a temporal segregation between target and competitor reactivations during the last retrieval repetition.

## Discussion

To remember a specific experience, the respective memory often needs to be selected against overlapping, competing memories. The processes by which the brain achieves such prioritisation are still poorly understood. We here tested a number of predictions derived from the oscillating interference resolution model (*Norman et al., 2006*). In line with its predictions, the present results demonstrate that the neural signatures of two competing memories become increasingly phase separated over time along the theta rhythm. Furthermore, larger phase differences were associated with fewer intrusion errors from the interfering memories. These findings support the existence of a phase-coding mechanism along the cycle of a slow oscillation that adaptively separates competing mnemonic representations, minimising their temporal overlap (*Lisman and Idiart, 1995*; *Norman et al., 2006*).

Based on the oscillating interference resolution model (*Norman et al., 2006*), we hypothesised that target memories would be strengthened and become decodable at an earlier (higher inhibition) phase (see *Figure 1d*), while competing associates would be weakened, with time windows of their reactivation gradually shifting to a later (low inhibition) phase. The phase separation between the two competing memories should thus increase the more often the competing memories are being reactivated, and we expected to find an observable difference in phase separation after a number of repeated recalls. To further test whether a large phase segregation is beneficial for memory performance, we related phase distance to the number of behavioural intrusion errors. Supporting our hypotheses, in the third and final recall repetition target and competitor memories showed a significant phase shift with an average distance of 34° (*Figure 4a and b*). Larger phase separation angles in individual participants were related to lower levels of behavioural interference. Participants with low levels of memory intrusions showed a significant phase difference of 57°, while high-intrusion participants showed a non-significant 7° phase shift on average (*Figure 4b*). Also in the last repetition, a significant phase modulation was found for both target and competitor memories such that the fidelity of their neural reactivation was rhythmically modulated by the dominant 3 Hz rhythm (*Figure 3e and f*). Finally, a significant cluster of cross-correlation of the continuous target and competitor decoding timecourses also suggested an average phase lag of 30°, complementing the analyses that explicitly relate classification performance to hippocampal theta phase. Though deviating from

the pre-registered hypotheses in some respects as discussed below, these results largely confirm our predictions derived from the oscillating interference resolution model (*Norman et al., 2006*), supporting a central role for slow oscillations in orchestrating memory recall by temporally segregating potentially interfering memories.

Based on the same model (*Norman et al., 2006*), we also predicted that across repeated retrieval attempts, target memories would be strengthened and competitors become less intrusive. Though not tested behaviourally in this study, repeated recall is known to induce enhancement of the retrieved memories (*Karpicke and Roediger, 2008*; *Rowland, 2014*) and forgetting of non-retrieved, competing memories (*Anderson et al., 1994*). Evidence for the up- and downregulation of neural target and competitor patterns, respectively, has previously been shown in fMRI studies (e.g. *Kuhl et al., 2007*; *Wimber et al., 2015*). In this work, we found neural evidence for both target strengthening and competitor weakening within the time window of maximum memory reactivation (*Figure 2h*). The increase in target evidence was more robust than the decrease in competitor evidence, with the latter not surviving multiple comparisons correction across the entire recall time window. It is possible that competitor decoding across all trials was too noisy in the first place, especially when focusing on trials where participants correctly answered both follow-up questions, and where neural patterns should thus be dominated by retrieval of the target memory. When instead analysing only incorrect trials, we found moderate evidence for the neural reactivation of competitor memories (*Figure 2—figure supplement 2d*). The relatively stable levels of competitor reactivation over repetitions might also be due to the lack of feedback in our task, allowing little adjustment of incorrect responses, as also reflected in the flat behavioural intrusion index (*Figure 2b*, *Figure 2—figure supplement 1a*). Overall, however, and in line with recent EEG work (*Bramão et al., 2022*), our results suggest that dynamic changes in overlapping memories can be tracked using MEG.

Decoding of target and competitor memories in the high-interference (CC) condition was generally less robust than decoding of target memories in the low-interference (NC) conditions. Generally, one challenge for time-resolved decoding of reactivated memory content is the considerable variance in the timing of memory recall across trials, conditions, and participants, likely affecting the timing in neural pattern reinstatement. Such variance can be rectified, at least to a degree, when aligning the timelines to the button press that indicates subjective recall, as shown by our response-locked analyses leading to more robust target decoding (see *Figure 2f and g*). There are, however, several reasons why we pre-registered our analysis locked to the onset of the memory cue. Most importantly, previous work suggests that the phase of theta oscillations is reset by a memory cue and remains relatively stationary for a period of time (see *Rizzuto et al., 2006*; *Ter Wal et al., 2021*), such that the locking of memory reactivations to the theta rhythm should be most coherent when aligned to cue onset. Note that all crucial analyses comparing phase shifts of target and competitor reactivations were conducted on the level of single trials. Therefore, these analyses are not affected by whether or not the reactivation maxima are consistent in time across trials and participants, even though such inconsistencies will dilute the average cue-locked decoding performance.

The functional relevance of slow oscillations in regulating the excitation–inhibition balance of specific neural assemblies has featured strongly in other computational models too, and these models are supported by our findings at various levels. Most relevant in the present context is the theta-gamma model (*Lisman and Idiart, 1995*; *Lisman and Jensen, 2013*), which assumes that the coupling of gamma bursts to specific theta phases provides a timing mechanism for ordering sequences of items in working memory. Here, a gamma burst reflects the firing of a local cell assembly representing a distinct unit of information, while the phase of the slower theta oscillation rhythmically modulates the excitability of these local neural assemblies, and feedback inhibition allows for an organised, non-overlapping firing sequence across the theta cycle. In humans, studies using episodic and working memory tasks have provided evidence for such a theta-gamma code (*Bahramisharif et al., 2018*; *Canolty et al., 2006*; *Fuentemilla et al., 2010*; *Griffiths et al., 2021*; *Herweg et al., 2020*; *Heusser et al., 2016*; *Karlsson et al., 2022*; *Reddy et al., 2021*; *Ten Oever et al., 2020*; *Ter Wal et al., 2021*). The oscillating interference resolution model (*Norman et al., 2006*) and theta-gamma phase-coding models share the fundamental assumption that slow oscillations regulate the excitation–inhibition balance of local neural assemblies (*Buzsáki, 2006*). However, they do differ in their theoretical scopes. Theta-gamma code models were developed to explain how the brain can handle ongoing states of high attentional or working memory load, where multiple distinct items need to be kept separated

and organised. The oscillating interference resolution model (*Norman et al., 2006*) is a learning model and thus explicitly oriented towards optimising future network states. It employs fluctuating inhibition to actively shape memory representations, with the goal to minimise future overlap and interference.

To our knowledge, no study in humans has explicitly tested for phase coding when several overlapping memories compete for retrieval. However, one intracranial study using a virtual navigation paradigm showed that the neural representations of potentially interfering spatial goal locations are coupled to distinct phases of the hippocampal theta rhythm (*Kunz et al., 2019*). Phase precession and theta coupling of neuronal firing during goal-directed navigation have also been shown in human hippocampal single-unit recordings (*Qasim et al., 2021*; *Watrous et al., 2018*). The present results directly support the idea that phase coding can temporally separate co-active, competing memory representations in the human brain. They additionally suggest that such a code adaptively evolves across repeated target retrievals, in line with the idea that oscillating excitation–inhibition can optimise learning with respect to future access to the relevant target memory.

A second prominent computational model focuses on different functional operations within the hippocampal circuit at opposing phases of the theta rhythm, with one half of the hippocampal theta cycle devoted to the encoding of new incoming information, and the other half to the recovery of internally stored information (*Hasselmo et al., 2002*; *Kunec et al., 2005*). In humans, previous EEG work *Kerrén et al., 2018* and a recent study investigating rhythmic modulations of reaction times *Ter Wal et al., 2021* have provided evidence for such process separation. In the present study, target and competitor reactivations peaked at distinct theta phases, with an average separation angle of approximately 35°, and 57° in the low-intrusion group that presumably reached optimal interference resolution. This observation is consistent with a framework where both targets and competitors are reactivated during the retrieval portion of a theta cycle (*Hasselmo et al., 2002*; *Kunec et al., 2005*), but with sufficient phase separation to allow for differentiation in time. Note, however, that the oscillating interference resolution model (*Norman et al., 2006*) dedicates the entire theta cycle to retrieval operations (as depicted in *Figure 1d*) and would thus predict temporal distances of up to 180° between targets and competitors. The two models may not necessarily be incompatible, especially when considering that encoding and retrieval computations take place in different subcircuits of the hippocampus (*Hasselmo et al., 2002*). Future modelling work is needed, however, to better understand the learning dynamics across the full theta cycle and reconcile the apparent theoretical conflict between the two frameworks.

Other predictions of the oscillating interference resolution model (*Norman et al., 2006*) relate to the neural overlap in spatial patterns and have been tested primarily in fMRI studies. One such prediction is the non-monotonic plasticity hypothesis, derived from the same learning dynamics along the oscillatory cycle as described above. If a competitor is sufficiently strong to be co-activated in the high-inhibition (target) phase, it will benefit from synaptic strengthening of common connections and thus be integrated with the target memory. Moderately co-active competitors, on the other hand, will be subject to synaptic depression and as a result become differentiated from the target memory (*Ritvo et al., 2019*). Evidence for non-monotonic plasticity comes from fMRI work that directly tracks changes in the similarity of target and competitor representations in terms of their spatial patterns (*Hulbert and Norman, 2015*; *Wammes et al., 2022*), and the representational changes dependent on the level of (co)activation of overlapping memories (*Detre et al., 2013*; *Lewis-Peacock and Norman, 2014*; *Poppenk and Norman, 2014*; *Wang et al., 2019*). Indirectly related, fMRI studies investigating representational change resulting from repeated learning or recall have also found evidence for differentiation or integration of overlapping memories in hippocampus, late sensory cortex, and prefrontal cortex (*Favila et al., 2016*; *Hulbert and Norman, 2015*; *Kim et al., 2017*; *Kuhl et al., 2010*; *Schlichting and Preston, 2015*; *Wimber et al., 2015*), and sometimes observed both in the same study but in different regions (*Molitor et al., 2021*; *Schlichting and Preston, 2015*). The present study was not designed to track representational changes, for example, in terms of comparing target and competitor representations before and after repeated recall. Arguably, however, there is a straightforward agreement between spatial and temporal separation in this network model. Strong competitors would tend to co-activate in the high-inhibition phase and thus be co-strengthened and integrated via LTP (*Bliss and Lomo, 1973*; *Hebb, 1949*), whereas moderately co-active competitor memories would tend to co-activate only at lower levels of inhibition, which according to the model would lead to LTD. Our data support the idea that low temporal separation is related to integration, behaviourally. The

high-intrusion group had a mean phase difference of 7° between target and competitor memories, theoretically in line with a time window of strong co-firing hence synaptic strengthening. Integration might thus have led to higher amounts of intrusions (*Brunec et al., 2020*). In the low-intrusion group, more pronounced phase separation between target and competitor memories (57° on average) might have led to only moderate co-firing and hence promoted differentiation, resulting in lower levels of behavioural interference. Although speculative, temporal separation could be a prerequisite for spatial differentiation. Future studies, combining high temporal with high spatial resolution, and paradigms to track the representational distance of overlapping memories, are needed to fully understand these dynamics.

A priori we included the entire theta band in our pre-registration as previous studies have indicated large variations in the peak theta frequency in humans (*Goyal et al., 2020*; *Jacobs, 2014*; *Kerrén et al., 2018*; *Lega et al., 2016*; *Ter Wal et al., 2021*). For example, in our previous study we found that reactivations of target memories were most consistently coupled to a hippocampal 7–8 Hz oscillation, even though a slower 3–4 Hz peak was also present (*Kerrén et al., 2018*). In another study, *Ter Wal et al., 2021* found that slower 2–3 Hz oscillations dominated behaviour and hippocampal signals specifically when responses depended on memory, consistent with other intracranial work reporting pronounced slow theta oscillations in the human hippocampus during memory tasks (*Goyal et al., 2020*; *Lega et al., 2012*). In working memory, it has even been demonstrated that theta oscillations slow down with increasing memory load (*Fell et al., 2011*; *Wolinski et al., 2018*), in line with the theoretical idea that slower rhythms accommodate better phase separation (*Lisman and Idiart, 1995*; *Lisman and Jensen, 2013*). Descriptively, such theta slowing was evident in the present data when comparing the frequency profiles of rhythmic memory reactivation between NC and CC conditions, where conditions in which only one associate needs to be remembered showed slightly faster rhythmic peaks (4–5 Hz) than the condition in which two associates compete for retrieval (3–4 Hz, see *Figure 3—figure supplement 1*). While offering a possible explanation why slower oscillations are often observed in tasks with high memory demands, this post-hoc interpretation will need to be corroborated by empirical evidence in future studies.

Phase coding offers an elegant theoretical solution to temporally segregate the processing of potentially interfering information. Much empirical evidence for phase sequencing along the hippocampal theta oscillation comes from spatial navigation work, both in rodents and humans (*Herweg et al., 2020*; *Kunz et al., 2019*; *O'Keefe and Recce, 1993*; *Reddy et al., 2021*). We here demonstrated that phase coding facilitates the separation of overlapping, associatively linked memories. Together with the computational model used to derive these predictions, these findings offer a possible mechanism utilised by the human brain to resolve competition between simultaneously active memories. More generally, they add to a growing literature showing that slow oscillations orchestrate the intricate timing of neural processing with direct, observable effects on behaviour.

## Materials and methods
### Subjects
Twenty-six right-handed participants (18 females, 8 males) took part for financial or course credit compensation (mean = 24.1 years, SD = 5.73 years, range 18–33). They all had normal or corrected-to-normal vision and reported no history of neurological disorders. All experimental procedures in this study were approved by and conducted in accordance with the University of Birmingham's STEM Research Ethics Committee (ERN-16-1512 and ERN-18-0226P). Written informed consent was obtained from participants before they took part in the experiment.

### Stimuli and task
The material consisted of 72 images depicting animate and inanimate objects (equal number of mammals, birds, insects, and marine animals, electronic devices, clothes, fruits, and vegetables), and 72 images depicting indoor and outdoor scenes, taken from the BOSS database (*Brodeur et al., 2010*) and online royalty-free databases. Stimulus selection was motivated by previous success at distinguishing these categories using time-resolved multivariate pattern analysis (*Kerrén et al., 2018*). Images from both object and scene classes were pseudo-randomly split for each participant into six sets, so that each set consisted of 20 objects and 20 scenes, 10 animate and 10 inanimate, 10 indoor

and 10 outdoor scenes. Each set constituted one learning block. Also, a list of 144 action verbs, largely overlapping with those used in *Linde-Domingo et al., 2019*, served as cue words in the cued recall task. These words were randomly assigned to images and conditions for each participant for creating the relevant word–image associations. Three additional associations were used for demonstrative purposes.

Participants received task instructions and first performed one short practice block. All participants (see exclusion below) then performed six experimental blocks (40 encoding trials and 72 (24 × 3) retrieval trials per block), each consisting of an associative learning phase, a distractor task, and a retrieval test with three repetitions per target item (*Figure 1*). At encoding, participants were asked on each trial to encode a word together with an image associate. In the CC, a word was encoded together with two associates, separated by at least three intervening trials. The instruction was to always memorise the most recent associate that was presented together with a given word for the subsequent memory test. Therefore, the second associate in the CC always served as the target, with the previously learned first associate (i.e. competitor) assumed to elicit proactive interference. In the non-competitive single-exposure condition (NC1), a word was encoded together with only one associate, and these associations were never repeated during encoding. This condition served as the behavioural baseline for measuring the effect of proactive interference on memory performance (i.e. having previously encoded a competing associate compared with only one associate). In the non-competitive double-exposure condition (NC2), participants also encoded a word together with only one associate, but these associations were presented twice. This condition served as the neurophysiological baseline and was specifically designed to control for neural effects induced by the repetition of the word cue (including but not limited to repetition suppression; *Epstein et al., 2008*; *Kristjánsson and Campana, 2010*). The conditions were shown in pseudo-randomised order within each block, where 1/3 of the associations were in NC1, 1/3 in NC2, and 1/3 in CC (NC1 = 8 associations, NC2 = 8 associations, CC = 8 associations). In NC1 and NC2, there were an equal number of objects and scenes (four objects and four scenes per block, but shown twice each for NC2, hence resulting in eight object and eight scene trials), whereas in CC there were eight object and eight scene trials, with one pair each linked to the same cue word. In total, this summed up to 40 trials for one block of learning (NC1 = 8 trials, NC2 = 16 trials, CC = 16 trials). The order of the trials belonging to the three conditions was pseudo-randomised such that the average serial position of each condition within a block was equal. Images were pseudo-randomly assigned to these conditions for each participant, with the constraint that in the CC the associates needed to be from different image categories (one object and one scene, split such that on half of the CC trials, the target was an object and the competitor a scene image, and vice versa for the remaining half).

A learning trial consisted of a jittered fixation cross (between 500 and 1500 ms), a unique action verb (1500 ms), a fixation cross (1000 ms), followed by a picture of an object or scene that was presented in the centre of the screen for 4 s. Participants were asked to come up with a vivid mental image that linked the image and the word presented in the current trial. As soon as they had a clear association in mind, they pressed the right-thumb key on the button box. Participants were aware of the later memory test.

A distractor task followed each learning phase. Here participants had to indicate whether a given random number (between 1 and 99) presented on the screen was odd or even. They were instructed to accomplish as many trials as they could in 45 s and received feedback about their accuracy at the end of each distractor block.

After the distractor task, participants' memory for the 40 verb–object associations memorised in the immediately preceding learning phase was tested in pseudo-random order, with three repetitions per relevant association. The order was pseudo-randomised such that all associations were first tested once in random order before testing all associations again in a new random order, and again a third time. No significant difference in serial position resulted when comparing CC and NC2 items [t(23) = –1.69, p=0.1], CC and NC1 items [t(23) = –1.18, p=0.25], or NC1 and NC2 items [t(23) = –0.43, p=0.66]. One repetition consisted of 24 trials (eight for NC1, eight for NC2, and eight for CC), with the constraint that a given association could only be tested once per repetition. Each trial consisted of a jittered fixation cross (500–1500 ms), followed by one of the words as a reminder for the association. Participants were asked to bring back to mind the most recent associate of this word as vividly as possible. The cue was presented on the screen for 500 ms and thereafter a blank screen with a

black empty frame was presented. To capture the particular moment when participants consciously recalled a specific association, they were asked to press the right-thumb key as soon as they had a vivid image of the associated memory in mind. If they did, the frame flashed once, and participants were presented for 4 s with a blank screen and asked to hold the image in mind. A question then appeared on the screen asking whether the retrieved item was an object, a scene, or they were unable to remember. Across trials, the object and scene options randomly shifted between the left and right sides of the screen. If the participant did not remember the association, they were told to press the left-thumb button. If participants selected 'object' or 'scene,' a follow-up question appeared (dependent on the response to the first question), asking whether the retrieved associate was an inanimate or animate object, or whether it was an indoor or outdoor scene. The two follow-up questions were self-paced, and there was no feedback.

The experiment was set up via custom-written code in MATLAB 2016a (The MathWorks, Munich, Germany) using functions from the Psychophysics Toolbox Version 3 (*Brainard, 1997*). The presentation was projected onto a screen located 1.5 m away from the participant using Windows 64 bit. Images were 2000 × 2000 pixels large.

## Analysis

The general hypotheses and analysis steps were pre-registered and can be found on OSF (here).

After excluding participants based on the criteria stated in the pre-registration (more than 2 SD from the mean accuracy for each condition separately), 24 participants (16 females, 8 males) remained, with an average age of 24.5 years (SD = 5.73). For the MEG analysis, a further 3 participants were excluded due to noisy data, which resulted in 21 participants included in the analyses. All participants performed all six blocks except for two, for whom time limit and button box errors occurred, resulting in only five blocks for these participants for MEG analysis.

## Statistical analysis

In view of the pre-registration on OSF, all statistics were one-sided and conducted on a group level. Non-parametric cluster-based permutation tests were conducted to test classification performance against chance across multiple time points, and for time–frequency analyses comparing CC and non-CC (see below). Only correct trials were used for all MEG analyses. Where a specific frequency band had to be preselected, the analyses were limited to 3–8 Hz. This, however, deviated from the frequency range of interest in the pre-registration, which was 4–8 Hz. Since the study was pre-registered, several studies have shown slower theta oscillation specifically related to human episodic memory (*Goyal et al., 2020*; *Ter Wal et al., 2021*) and we therefore decided to include also 3 Hz in the frequency band of interest.

## MEG data analysis

The MEG was recorded at the Centre for Human Brain Health (CHBH), Birmingham, UK, using an Elekta Neuromag TRIUX system, with 306 channels (204 planar gradiometers and 102 magnetometers; only gradiometers are used for all analyses reported here), sampled at 1000 Hz (Elekta, Stockholm, Sweden). EEG was recorded with a 64-channel electrode cap in the initial 10 participants as a sanity check in order to verify that a strong mid-frontal theta signal can be observed in the CC > NC2 condition, and how its topography compares between MEG and EEG (*Cavanagh and Frank, 2014*). Since we found a comparably strong theta increase over EEG and MEG (gradiometer) sensors, the EEG data is not reported in any of the analyses presented here. The experiment was shown on a projector screen using a PROPixx projector (VPixx Technologies, Saint-Bruno, Canada) with a 1440 Hz refresh rate, and participants' responses were collected using two button response boxes (fMRI Button Pad [2-Hand] System, NAtA Technologies, Coquitlam, Canada).

For source localisation, individual anatomical MRI scans (T1-weighted; 1 × 1 × 1 mm voxels; TR = 7.4 ms; TE = 3.5 ms; flip angle = 7°, field of view = 256 × 256 × 176 mm) were acquired at the CHBH (3T Achieva scanner; Philips, Eindhoven, the Netherlands).

## Preprocessing

Preprocessing was done using the FieldTrip toolbox versions 2019 and 2021 (*Oostenveld et al., 2011*) and custom-written MATLAB code. A Butterworth low-pass filter of 200 Hz and a band-stop

filter (50 Hz, 100 Hz, and 150 Hz) were applied to the data. To avoid temporal displacement of information, while still removing slow drifts, trial-masked robust detrending was applied to the retrieval data, in accordance with recent recommendations (*van Driel et al., 2021*). Briefly, the data for this correction step were divided into epochs spanning 15 s before cue onset and 15 s after cue onset. The experimentally relevant events started 1000 ms before cue onset until 4000 ms after cue onset. This time window was masked out from each trial. To make sure all data would be included, the continuous data were symmetrically mirror-padded with 15 s prior to segmentation. To improve the fit of the higher-order polynomial, a first-order polynomial was used to detrend the entire epoch (in accordance with *de Cheveigné and Arzounian, 2018*). Thereafter, a 30th-order polynomial was fitted and removed from the data. Note that the events of interest were not a part of the fitting procedure, making sure the fit was not being influenced by cognitively relevant processing. The method was partly implemented using the Noise Tools toolbox (http://audition.ens.fr/adc/NoiseTools) together with custom-written MATLAB code. After this step, the detrended data were cut into the experimentally relevant epochs (−1000 to 4000 ms around cue onset at retrieval).

An automatised trial and component rejection was applied to the data. In a first step, to remove high-frequency bursts, data were high-pass filtered at 100 Hz, and trials that exceeded four times the median absolute deviation of the amplitude distribution across trials were automatically removed. In a second step, independent component analysis (ICA) was used to detect artefacts to be removed in the data. To this end, MEG data were downsampled to 250 Hz, and only the first 1.5 s after cue onset were used for ICA (to reduce computational load). The unmixing matrix was then applied to the entire epoch of non-downsampled data. To remove blink and cardiac artefacts, a template was created by running an ICA on the six first participants. The components for the two artefacts were then averaged separately, and each average component was used as a template to identify matching components in each participant. For the blink template, the algorithm correlated its topography with that of each component identified in an individual participant, and subsequently removed the component with the best match. For the cardiac component, which is difficult to identify based on topography, the algorithm filtered the component time series to between 10 and 20 Hz and correlated the peak amplitude of each trial with the cardiac time-series template. Any component which correlated to a greater extent than four times the median absolute deviation of the correlation was classed as a cardiac component and removed. All removed components were visually inspected for validation.

After these components were removed, an additional automatised artefact rejection was conducted, similar to the initial step. The same procedure as in the first rejection round was followed but was done on channels instead of trials. Again, data were band-pass filtered between 1 and 100 Hz, and the channels that exceeded three times the median absolute deviation of the channel distribution were rejected. We chose three times the median absolute deviation for the last two rounds of rejection to be more conservative. Bad channels were interpolated using the triangulation method (*Oostenveld et al., 2011*). After components and trials were removed, all trials and components were again visually inspected, and trials and components still containing artefacts were manually removed. Lastly, data were downsampled to 250 Hz. Due to poor data quality in 3 participants, remaining data from 21 participants were used for MEG analyses. On average, 353 out of 432 retrieval trials were kept (min = 290, max = 397, SD = 27) for analysis. In the CC, the number of trials on average was 118 (max = 140, min = 95, SD = 10.24). Per retrieval repetition there was an average of 40 (SD = 3.81), 39 (SD = 3.63), and 38 (SD = 4.31) trials for each respective repetition. For the NC1 and NC2 collapsed, the average number of trials per retrieval repetition were 79 (SD = 7.38), 77 (SD = 6.36), and 77 (SD = 7.29). At the end of preprocessing, a sanity ERF-check on occipital channels of the retrieval data was conducted and the average waveform can be found in *Figure 3—figure supplement 2a*.

## Time–frequency decomposition

The spectral difference between the CC and neural baseline (NC2) during retrieval was calculated by convolving the combined activity of each planar channel (summing the magnitude of the planar gradient over both directions at each sensor) with a complex Morlet wavelet of a minimum of five cycles (increasing with frequency to cover approximately 500 ms in length [e.g. 10 Hz: 10 * 0.5 = 5 cycles]) from 1 to 20 Hz for each condition. First, paired-samples *t*-tests were computed between the two conditions to investigate the difference in spectral power. To account for multiple comparisons across time points (0–1000 ms post-cue), frequencies (3–8 Hz), and sensor (102 combined gradiometers), the

*t*-statistics were subjected to non-parametric cluster-based permutation testing, as implemented in the FieldTrip software. The threshold for statistical testing was set to a cluster alpha level of 0.05. The minimum number of neighbouring channels that were considered a cluster was set to three. *t*-Values above the threshold of 0.1 were then summed up in a cluster and compared against a distribution where condition labels were randomly assigned 1000 times with the Monte Carlo method, following the default method implemented in FieldTrip.

## Multivariate pattern analysis

Two separate LDA-based classifiers were trained on discriminating subordinate picture class, that is, animate vs. inanimate for objects, and indoor vs. outdoor for scenes. This important feature of the design allowed us to have two classifiers that provide independent measures of target and competitor reactivation. The two classifiers were trained on the non-CCs ('pure') (i.e. retrieval of objects and scenes in the NC1 and NC2 conditions), and were then tested on both the non-CCs and CCs. The time interval of interest for these multivariate analyses started 500 ms before onset of the retrieval cue and lasted up to 3000 ms post-cue. This time window was selected because no memory reactivation is expected earlier than 500 ms (*Staresina and Wimber, 2019*), and previous work using a similar (though non-competitive) cued recall paradigm suggests that participants need approximately 3 s to mentally reinstate a image (*Linde-Domingo et al., 2019*; *Kerrén et al., 2018*). A sliding Gaussian window with a FWHM in the time domain of 40 ms was applied to the trial time series before classification analysis. Each retrieval trial was then baseline corrected by subtracting the mean amplitude in a time window from 400 to 50 ms pre-cue, separately per sensor. Training and testing were conducted time point per time point, in steps of 8 ms, each time point centred at the centre of the Gaussian smoothing window. The MEG gradiometer sensor patterns (amplitudes on each of the 204 channels at a given time point) were used as classifier features, independently per participant.

Training/testing in the non-CC was done as a sanity check for classifier performance in a memory retrieval situation with no interference. Since the same data was used for training and testing in this case, a tenfold cross-validation was used and repeated five times. To test for reactivation of target and competitor memories in the CC, trials were split into those where the target was an object and the competitor was a scene, and vice versa, and the corresponding object and scene classifiers (trained on non-competitive retrieval) were then used to separately indicate evidence for target and competitor reactivation on each single trial. Only correct trials were used in the CC. As the training and testing data came from different trials, cross-validation in the CC was not necessary. To avoid overfitting, the covariance matrix was regularised using shrinkage regularisation, with the lambda set to automatic (*Blankertz et al., 2011*).

The LDA, as used here, reduces the data from 204 channels into a single decoding timecourse per trial, and we used these single-trial, time-resolved decision values (d-values or fidelity values) of the classifier as an index of memory reinstatement (*Carlson et al., 2014*). Classifier accuracy was derived by calculating the fraction correctly predicted labels, whereas chance was defined as 50% for a binomial classifier. More specifically, during training, the classifier found the decision boundary that could best separate the patterns of activity from the two classes (animate vs. inanimate for objects, indoor vs. outdoor for scenes) in a high-dimensional space. The classifier was then asked to estimate whether the unlabelled pattern of brain activity in any given retrieval trial and at each time point was more similar to one or the other class. This training test procedure was repeated until every single retrieval trial had been classified. A larger smoothing kernel of 200 ms FWHM in the time domain was applied to the decoding timecourses purely for visualisation but not for statistical analysis purposes.

As part of a (not pre-registered) set of analyses conducted in response to peer review, we also analysed decoding performance after realigning the data to the button press that indicates subjective recollection of the target associate. These analyses followed the same approach as above, except that we excluded trials in which responses occurred in the first 500 ms or last 200 ms to be able to plot decoding accuracy from −500 to +200 around the button press.

## Determine peak frequency of fidelity values using IRASA

The IRASA method has been shown to robustly find and separate the oscillatory from the fractal signal both in ECoG and MEG data, and was here used to quantify the oscillatory signal component of the fidelity values (*Wen and Liu, 2016*). More specifically, the brain produces task-related rhythmic

(oscillatory) components, but also arrhythmic scale-free (fractal) components (**Buzsáki and Draguhn, 2004**). The rhythmic oscillatory components are regular across time, whereas the fractal components are irregular (**Wen and Liu, 2016**). In short, IRASA resamples a time-series signal and computes a geometric mean of every pair (oscillatory and fractal) of the resampled signal. The median of the geometric mean is then used to extract the fractal power spectrum. The difference between the original power spectrum and the fractal power spectrum is the estimate of the power spectrum of the oscillatory component of the signal (**Wen and Liu, 2016**). In this study, IRASA (as implemented in the FieldTrip toolbox) was applied to the fidelity values, in a time window from 500 to 3000 ms post-cue, padding each trial length up to the next power of 2. Apart from the reasons mentioned above, this large time window also assures that low frequencies can be properly estimated.

To test for significance in the frequency-transformed decoding timecourses, the LDA as described above was repeated 15 times per subject, now trained on random labels in each iteration. The fidelity values from each iteration were then subjected to the same IRASA method as the non-shuffled data. An empirical null distribution was created on the second (i.e. participant) level as follows. In each of 1000 repetitions, one of the 15 surrogate-label classifier outputs (i.e. the oscillatory component output by IRASA) per participant was randomly selected, and an empirical group average was computed for this random selection. This procedure resulted in 1000 group averages of oscillatory strength estimates for each given frequency of interest, representing the chance distribution at this frequency. The IRASA output of the real-label classifier could then be contrasted against this chance distribution (**Cohen, 2014**). Note that due to the algorithm for separating the fractal components from the oscillatory components, the output from IRASA yields a higher frequency resolution than 1 point per frequency. However, our a priori frequency range was set to 3–8 Hz, and we therefore tested for significant oscillatory components in each 1 Hz frequency bin between 3 and 8 Hz, with the estimated chance distribution subtracted from the real value and subsequently divided by the standard deviation of the estimated chance distribution (**Figure 3a and b**). This gave a z-value, which was compared to the critical threshold of z = 2.32 at p=0.01, correcting for five multiple comparisons (3–4 Hz, 4–5 Hz, 5–6 Hz, 6–7 Hz, and 7–8 Hz). Again, this procedure replicated the one used in **Kerrén et al., 2018**.

## Source analysis

The aim was to extract activity from hippocampal virtual channel, hence, the magnetometers were used and their data projected into source space using an LCMV beamforming approach (**Gross et al., 2007**). Individual MRI scans (anatomical data) were available for 18 participants, and for the remaining 3 participants the standard head model as implemented in FieldTrip was used. Anatomical scans were aligned based on the sensor position obtained from a Polhemus system (Colchester, Vermont, USA). This was done in three steps: the first step was done manually by adjusting the alignment between the anatomical data and the head position; the second was done by the Iterative Closest Point algorithm implemented in the FieldTrip toolbox **Oostenveld et al., 2011**; and the third step was again done manually to check that the alignment worked, and adjusted if it did not. The realigned model was used to reslice and segment the brain to make the axes of the voxels consistent with the head position and subsequently to extract the brain compartments. The segmented brain was then used to create a forward model (head model). In our case, we used a semi-realistic forward model (**Nolte, 2003**). The forward model was used to create the source model (lead field), where for each grid point the source model matrix was calculated with a 1 cm resolution, and the virtual sensors were placed 10 mm below the cortical surface, and subsequently warped into each brain. In total, we modelled 3294 virtual sensors for each participant with whole-brain coverage. LCMV beamforming was used to reconstruct the activity of all virtual channels in source space (see below for selecting hippocampal channels) with 20% regularisation. To confirm that the source localisation provided reliable results, we checked sources for the early visually elicited response to the cue word in an early time window by calculating the pre-/post-variance around cue onset (–200 ms to 0 and 0 to 200 ms). Because of arbitrary orientation of the dipole in source space, variance is an unbiased measure to investigate where in space and when in time activity fluctuates (see **Figure 3—figure supplement 2b**).

For the phase-amplitude coupling between source-localised hippocampal 3 Hz phase and fidelity values, the filters were computed on 4 Hz low-pass-filtered data, from –1000 ms to 3000 ms after cue onset (baseline-corrected [–400 to –50]). The filters were obtained from correct trials in the competitive condition. The left and right hippocampus were chosen as a region of interest, derived from the

AAL atlas (*Oostenveld et al., 2011*). The individual MRI and the atlas were interpolated using the interpolation method *nearest*, finding the nearest region in the AAL atlas based on the Euclidean distance. Single-trial timecourses were extracted from the corresponding virtual sensors, and activity averaged across those sensors in the left and right hippocampus, resulting in one activity timecourse per trial for left and right hippocampus, collapsed.

Further source-level analyses were conducted to estimate the frequency profile of the raw trial data in the hippocampal region of interest and compare it to control regions (superior occipital cortex and precentral gyrus), with the hypothesis that the hippocampus would show a stronger power in the theta frequency range (3–8 Hz) than other regions. We used IRASA on each trial and each virtual channel, following the same procedure as we did when calculating the frequency profile of the fidelity values. For statistical comparison of the hippocampal profile against each of the two control regions, we directly contrasted the average power in our predefined 3–8 Hz frequency window of interest using two paired-samples *t*-test. Tests were conducted one-tailed because we expected higher theta power in the hippocampus compared to each of the other two regions. Since we conducted two separate tests (one per control region), we thus set the Bonferroni-corrected p-threshold for each test to.05.

## Phase-amplitude coupling between MEG data and fidelity values

The MI (*Tort et al., 2010*) was calculated to test for a relationship between the fidelity values and a hippocampal theta oscillation, with 3 Hz selected as the frequency of interest based on the results from previous steps (IRASA, see above). The signal from virtual channels was transformed to an analytic signal by convolving the raw data with a complex Morlet wavelet with cycles increasing with frequency to be approximately 500 ms in length for each frequency as implemented in FieldTrip (*Oostenveld et al., 2011*). Each complex value data point was then point-wise divided by its magnitude, resulting in a 4D matrix of phase values, containing trials * channels * frequencies * time. Phase values at 3 Hz were divided into 10 adjacent bins, ranging from – pi to pi, following the pre-registered procedure. To link classifier-based memory reinstatement indices to the hippocampal phase, the amplitudes of the single-trial fidelity values from corresponding time points were sorted into their respective phase bins, and the average classifier amplitude of each phase bin was calculated.

Following this binning procedure, two metrics were of relevance: (1) the MI showing the extent to which the fidelity values are modulated by the hippocampal phase, and (2) the maximum bin for target and competitor reinstatement. The MI was obtained by creating a uniform distribution and calculating the Kullback–Leibler distance and subsequently the MI, in accordance with the approach originally proposed by *Tort et al., 2010*. To calculate the pairwise phase difference between the target and competitor reinstatement, the absolute value of the complex number (the real and the imaginary part) representing the average maximum reinstatement peak for each participant was obtained by taking the square root of the sum of the squares of the parts (using the Pythagorean theorem). The amplitude was arbitrarily set to 1 because we were only interested in phase angle. The phase vectors were then point-wise divided with the complex modulus. Finally, to subtract the angles from each other, one vector was rotated by multiplying it with the other's complex conjugate. To obtain the angle, the inverse tangent of the ratio was taken of the product between the vectors. In MATLAB, the following code was used:

$$\theta = angle \left( \exp \left( 1i \times X \right) . \div \exp \left( 1i \times Y \right) \right)$$

where θ is the vector of angle difference.

We used the CircStat toolbox (*Berens, 2009*), as implemented in MATLAB, to statistically test the phase difference between target and competitor memories. More specifically, to test the difference in each repetition, as well as the average distance collapsed across all repetitions, we used a Rayleigh test for non-uniformity (*circ_rtest* function in the toolbox). This test will show significant clustering if the phase differences between target and competitor memories are non-uniform, independent of their absolute mean angle. To test whether the mean phase difference we obtained was significantly different from zero phase shift, we used the *circ_vtest* function with a specified mean angle of 0. This test indicates whether the mean angle of the phase distance is significantly different from zero. Lastly, we wanted to test the difference between the high- and low-intrusion groups' mean angle. To do so, we calculated the mean phase angle of target and competitor reactivation for each participant. We then calculated the circular mean of the two vectors within each participant, where the length of this

vector defines the similarity between target and competitor phase angles, independent of absolute angle. This vector is non-circular and can be subjected to a Wilcoxon signed-rank test. We chose a Wilcoxon signed-rank test as the sample size was low in each subgroup (n = 10 in each group, removing one participant from the larger group randomly to equate group size).

## Cross-correlation and maximum phase difference between fidelity timecourses of target and competitor

As another part of the set of new analyses conducted in response to peer review, we wanted to establish that the phase shift between target and competitor reactivation can also be observed when using the continuous fidelity timecourses, rather than zooming in on the reactivation peaks only. Single-trial fidelity timecourses from the third recall repetition (correct trials only), as used for the above phase-to-classifier locking analyses, were filtered at 3 Hz, and their lag was quantified using a cross-correlation. A sliding-window approach was used to preserve some degree of time information and investigate whether phase lag between signals evolves over time (within trials). Each sliding window had a length of 330 ms (1 divided by 3 Hz), allowing us to express the lag in phase angles from –180 to 180°. For each participant, 25 surrogates were computed where in each iteration, decoding was repeated 25 times, trained on random labels. The same cross-correlation procedure was then conducted on these surrogate classifiers. To obtain a z-value, the surrogate data was subtracted from the real data and subsequently divided by the standard deviation of the surrogate data. Non-parametric cluster-based permutation testing, as implemented in the FieldTrip software, was used to account for multiple comparisons across time points (500–3000 ms) and phase angle (lag: –180 to 180°). The threshold for statistical testing was set to a cluster alpha level of 0.05. t-Values above the threshold of 0.05 were then summed up within a cluster and compared against a distribution where condition labels were randomly assigned 1000 times with the Monte Carlo method, following the default method implemented in FieldTrip.

To further corroborate a phase shift between the average target and competitor fidelity timecourses, we filtered the single trials from the third repetition at the relevant 3 Hz frequency. We then obtained the phase using a Hilbert transform. To statistically quantify the phase shift on a group level, we subtracted, for each time bin, the phase angle of the target from the angle of the competitor (in complex space) for each participant's averaged fidelity value using the same method as described in the section 'Phase-amplitude coupling between MEG data and fidelity values'.

## Acknowledgements

This work was supported by a fellowship from Stiftelsen Olle Engkvist Byggmästare awarded to MW and CK and a Starting Grant from the European Research Council awarded to MW (ERC-2016-StG-715714). We also thank Sebastian Michelmann for helpful conceptual input during data analysis.

## Additional information

### Funding

| Funder | Grant reference number | Author |
| --- | --- | --- |
| European Research Council | 2016-StG-715714 | Maria Wimber |
| Stiftelsen Olle Engkvist Byggmästare | Awarded for Ph.D studies | Casper Kerrén Maria Wimber |

The funders had no role in study design, data collection and interpretation, or the decision to submit the work for publication.

### Author contributions

Casper Kerrén, Conceptualization, Resources, Data curation, Software, Formal analysis, Funding acquisition, Validation, Investigation, Visualization, Methodology, Writing - original draft, Project administration, Writing - review and editing; Sander van Bree, Benjamin J Griffiths, Formal analysis,

Methodology, Writing - original draft; Maria Wimber, Conceptualization, Data curation, Supervision, Funding acquisition, Validation, Methodology, Writing - original draft, Project administration, Writing - review and editing

## Author ORCIDs
Casper Kerrén ![ORCID] http://orcid.org/0000-0003-4870-6072
Benjamin J Griffiths ![ORCID] http://orcid.org/0000-0001-8600-4480
Maria Wimber ![ORCID] http://orcid.org/0000-0002-1917-353X

## Ethics

Human subjects: All experimental procedures in the present study were approved by and conducted in accordance with the University of Birmingham's STEM Research Ethics Committee (ERN-16-1512 and ERN-18-0226P). Written informed consent was obtained from participants before they took part in the experiment.

## Decision letter and Author response

Decision letter https://doi.org/10.7554/eLife.80633.sa1
Author response https://doi.org/10.7554/eLife.80633.sa2

---

# Additional files

## Supplementary files
- MDAR checklist

## Data availability

All data and code used in this project are made available on OSF and Zenodo.

The following datasets were generated:

| Author(s) | Year | Dataset title | Dataset URL | Database and Identifier |
| --- | --- | --- | --- | --- |
| Kerrén C, van Bree S, Griffiths BJ, Wimber M | 2022 | Phase separation of competing memories along the human hippocampal theta rhythm | https://doi.org/10.5281/zenodo.6602054 | Zenodo, 10.5281/zenodo.6602054 |
| Kerrén C, van Bree S, Griffiths BJ, Wimber M | 2022 | Target and competitor reactivation in episodic memory | https://osf.io/evgaw/?view_only=fbb676ccb2e74ccbb16d5d8aa8f9c58f | Open Science Framework, 10.17605/OSF.IO/EVGAW |

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
