## [Editor Report]

This pre-registration study used a proactive interference task in combination with MEG recordings on humans to test predictions of a previous computational model postulating that neural representations of competing memories are associated with varied phases of hippocampus theta-band rhythm. Their results confirmed the hypothesis and suggest that reactivations of target and competitor memories indeed occur at different phases of theta oscillations, which is further related to the intrusion effect in behavior.

---

## [Decision Letter]

**Decision letter after peer review:**

Thank you for submitting your article "Phase separation of competing memories along the human hippocampal theta rhythm." for consideration by *eLife*. Your article has been reviewed by 3 peer reviewers, one of whom is a member of our Board of Reviewing Editors, and the evaluation has been overseen by Laura Colgin as the Senior Editor. The following individual involved in the review of your submission has agreed to reveal their identity: Ehren L Newman (Reviewer #2).

All the 3 reviewers are impressed by the preregistration study, which makes the hypothesis-testing clear and well-grounded. They also appreciate the large consistency between the results and the tested hypotheses and concur that the findings would be of broad interest to memory and learning fields. Meanwhile, they have also raised several major concerns about the results (see major points below and specific comments by each reviewer).

Essential revisions:

(1) Concerns were raised about the lack of significant memory reactivations for the CC condition, even at the 3rd repetition when memories are supposed to increasingly be reactivated. The authors collapsed target and competitors (Supp Figure 3B) aiming to support memory reactivations, but the rationale is unconvincing since target and competitors are claimed to occur at different phases and should not be combined for decoding.

(2) Related to the 1st point, by comparing reactivations of the 3rd and 1st repetition, the authors suggest a reactivation increase over repetitions. The authors could also consider comparing the 3rd repeat and chance level to examine the memory reactivation at the 3rd repetition. Moreover, how do they reconcile the different reactivation latencies for CC and NC conditions?

(3) Regarding the theta rhythm in classifier fidelity and its relationship to midline theta-band signal, the current results support the phase separation only at the 3rd repetition. It would be important to see how the rhythm of classifier fidelity changes over repeats as well as its characteristics for the NC condition. Moreover, is there any way to confirm that the theta-band rhythm emerges from the hippocampus?

(4) There are other behavioral analyses that could be done to relate the neural findings to the behavior in addition to the intrusion effect (for example: probability of intrusion, RT for memory recall, overall accuracy, etc.). Please see details below in the reviewers' comments.

(5) The authors could consider adding representative data examples to illustrate the phase separation findings.

*Reviewer #1 (Recommendations for the authors):*

1. The decoding performance didn't reach statistical significance for both target and competitor (Figure 2e), which makes me skeptical to what extent the decoding courses truly denote the memory reactivations. If not, how could the phase-based memory reactivation conclusion be supported? I noticed that the effect mainly occurred at the 3rd repetition recall. Thus, I would recommend that the authors should at least provide evidence that the decoding performance was significant for the target and competitor in the 3rd repetition.

2. The target and competitor showed similar profiles instead of an out-of-phase manner (Figure 1d). Their average decoding performance even showed significant reactivation (supplemental figure 3b). The results seem to be not consistent with the major phase separation findings. Moreover, to flesh out the phase separation results, I would recommend the authors provide typical data examples from their results explicitly illustrating the phase separation profiles, e.g., a representative subject, etc.

3. The key hypothesis is that the neural representation of the target would become stronger, and phase separation would become more prominent across repetitions. They did find stronger reactivation and phase separation for the 3rd repetition compared to the 1st repetition. Meanwhile, is there any corresponding behavioral evidence such that the overlapping memories become less distracting across repetitions?

4. There is no test to confirm the significant theta rhythm in the hippocampus detected in the present study. I think the authors should provide neural evidence backing up that the theta-band rhythm analyzed in their study indeed derives from the hippocampus.

*Reviewer #2 (Recommendations for the authors):*

I found much to appreciate about this manuscript and work. Much of that is summarized in my 'Public review' so I won't repeat it here.

I had no 'substantial concerns' about the quality of the work or interpretations presented. That is to say, none of the following thoughts that I'll share next should be seen as show-stopping. I will share these thoughts nonetheless in case they can support the growth of this work.

– The reasons why the reactivation effects should only become visible in the 3rd repetition were not clear to me. I struggle to come up with a just-so story based on the Norman et al. 2006 model or otherwise to understand these. Some assistance with this could help.

– There is a strange discrepancy in the frequency band of the frontal midline theta (~7 Hz) and the frequency of the phase modulated reactivations. I did not see any attempt to reconcile this. Are they simply two totally different things, the 7Hz, and 3Hz thetas?

– There are other tests that could be done to relate the reactivation dynamics to the behavior than are described. For example, one could compare the degree of reactivation to the probability of intrusion. Newman et al., 2010 – https://pubmed.ncbi.nlm.nih.gov/20181622/ – for example did something similar. At the coarsest level, the probability of an intrusion should be lowest for trials where there was no evidence of competitor reactivation.

*Reviewer #3 (Recommendations for the authors):*

There are some suggestions that could improve the overall conclusions one can draw from the manuscript.

The manipulation of repeating the retrieval period is interesting and allows for some novel hypotheses and questions. The authors use a measure of intrusions (by assessing how often subjects select the competitor subordinate category) and find that this is independent of the recall trial number. How about accuracy for the CC conditions?

The authors find an increase in fontal theta that lateralizes to the right for the competitive condition (CC). While the authors find the lateralization surprising, and instead expected a greater increase over midline structures, the lateralization here may be more consistent with recent literature implicating the right DLPFC in action inhibition. In this case, the inhibited action may be related to the competitor's memory. Alternatively, if there is a conflict signal that is relevant for this task, although the authors look for this conflict signal during retrieval it may also be helpful to identify whether a conflict signal is present during the encoding portion of the task when the competitor memory is introduced.

One of the advantages of the task design is that the subordinate categories allow for classifiers to be built that can decode which memory is being reactivated. The authors use an LDA-based classifier on the MEG sensor amplitudes to construct and test the classifiers. Interestingly, when decoding the retrieval data, the classifiers are significant in the NC conditions a full 2.5 seconds after the cue. This seems like a very long time, as the authors acknowledge. In the description of the task, the authors report that the subjects indicate when they have the picture in mind, and then indicate the supra- and subordinate categories. Presumably, then, this means there is a response time. How does this activation compare to the response time?

The authors report that there is no significant reinstatement of the target category during correct competitive trials as compared to the competitor category, although they claim that the classifier performance peaks at a similar time to the NC condition. This seems a bit concerning because one would expect to see some evidence of reactivation if subjects are making the correct decision. How should this be reconciled? Instead, the authors find that the evidence for target memory increases over recalls, which they offer as evidence that these memories are increasingly being reactivated. However, classifier performance peaks at a different time than when you see reinstatement in NC condition and is in fact earlier. Why?

It would be helpful if the authors could please clarify Supp Figure 3B. What does it mean when they collapse targets and competitors? Do they mean that the classifier can decode up either one?

If the authors are seeing increasing evidence of target information over several repeats, then this raises the question as to whether the classifier of subordinate target category would only work if only looking at the third repeat? The authors test this by comparing classifier performance in the 1st v 3rd repeat. However, how about comparing performance in the 3rd repeat versus chance?

The authors then tie in classifier decoding with the phase of the theta rhythm in the hippocampus. First, the classifier fidelity itself appears to have a 3 Hz rhythm. They then compute the modulation index (MI) of classifier fidelity to the hippocampal 3Hz phase. Classifier fidelity is only modulated by the hippocampal phase during the 3rd retrieval repetition. Similarly, we only see significant phase differences between target and competitor in the last repetition. This would seem to support the hypothesis that repetition leads to greater separation and better memory. However, it would be then helpful to know how the rhythm of classifier fidelity (the underlying 3Hz rhythm of classifier performance) changes across repeats. Is this fixed, or does this also exhibit changes with repetition? Is this also the case in the NC condition? For the NC condition, how do the phases modulation compare with the phases of target and competitors in the CC condition?

---

## [Author Response]

Essential revisions:(1) Concerns were raised about the lack of significant memory reactivations for the CC condition, even at the 3rd repetition when memories are supposed to increasingly be reactivated. The authors collapsed target and competitors (Supp Figure 3B) aiming to support memory reactivations, but the rationale is unconvincing since target and competitors are claimed to occur at different phases and should not be combined for decoding.

We agree that evidence for memory reactivation in the critical CC condition was weak in the previously reported (and preregistered) analyses. We added new analyses showing that robust memory reactivation is present in the CC condition when realigning the timelines to the response (i.e., time point of subjective recollection), rather than the cue onset, which we hope will reassure the readers and reviewers of the validity of our decoding algorithm and approach. We also clarify that the collapsed accuracy for target/competitor decoding only served the particular purpose of identifying a common time window of reactivation.

Regarding the first point of low decoding accuracy in the CC, we reasoned that there is considerable variability in the timing of memory reactivation across trials and participants that will make it difficult to see clear peaks of decoding accuracy when averaging across participants. The revised manuscript now includes several analyses where we realigned decoding to the button press that subjects made on each trial to indicate they recalled the associated memory. This approach (see Author response image 1) revealed significant clusters of classification accuracy preceding the button press by approx. 200ms. In Author response image 1, we also show that target classification on correct trials is significantly stronger than on incorrect trials, indicating that our method of detecting memory reinstatement is sensitive to behavioural memory success. Lastly, we also find evidence (though not surviving cluster-correction) of competitor reactivation when only using incorrect trials where the competing memory is more likely to dominate, both when doing the analysis response-locked (Author response image 1) and cue-locked (Figure 2 – supplement 2D; Author response image 1), the former again in a similar time window around 200ms before response.

**Author response image 1. sa2fig1:** Decoding performance time-locked to the subjective recollection button press. We realigned the trials based on response a. We find significant target decoding when averaging over all repetitions for targets in the competitive condition (CC), p_cluster_ <.05. b. A similar pattern is evident when contrasting target decoding between correct and incorrect trials in the CC (p_cluster_ <.05). These results reaffirm that the lack of significant decoding in the cue-locked analyses is due to timing differences in memory reinstatement between trials and participants, which is rectified when locking to the time point of subjective recollection. c. When only analysing incorrect trials, we find evidence of competitor decoding in a similar time window as when target memories were maximally reactivated in (a and b) (p_uncorrected_ = .05). d. Lastly, analysing competitor decoding on all incorrect trials from cue onset, we again found evidence for competitor reactivation in an early and a later time window (p_uncorrected_ = .05).

It is important to emphasise that we preregistered our decoding analyses locked to cue onset in order to maximise our chances to observe theta-locked memory reactivations. Previous work showed that the phase of theta oscillations is reset by a memory cue (Rizzuto et al., 2006; ter Wal et al., 2021), suggesting that the locking of memory reactivations to the theta rhythm will be most coherent for a period of time following cue onset. Since we conducted the crucial analyses of classifier-to-theta locking on the level of single trial fidelity values, comparing phase shifts of target and competitor, the analyses will not be affected by whether or not these reactivation peaks are consistent in time across participants, as they would need to be to reveal robust average cue-locked decoding performance.

Regarding the analysis collapsing of target and competitor decoding, as seen in Supp Figure 3B [now Supp Figure 2E], we understand that this point needs clarification. This analysis was conducted purely for the purpose of obtaining an unbiased time window (i.e., not biased either target or competitor decoding, as suggested as good practice in Cohen (2014)) in which further analyses on the up- and down-regulation of the two memories across repetitions could then be performed (shown in Figure 2 – supplement 2E and Figure 2H). We clarify this rationale in a sentence on p.49 l. 8-12.

For all other analyses, target and competitor decoding was kept separate.

(2) Related to the 1st point, by comparing reactivations of the 3rd and 1st repetition, the authors suggest a reactivation increase over repetitions. The authors could also consider comparing the 3rd repeat and chance level to examine the memory reactivation at the 3rd repetition. Moreover, how do they reconcile the different reactivation latencies for CC and NC conditions?

We followed this suggestion and compared the third repetition of target decoding with chance (50%). The analysis revealed a significant cluster emerging from 1.81 to 2.08 seconds after cue onset. However, this cluster did not survive a more stringent cluster-based permutation correction for multiple comparisons across time (p_corr_ = .07). We added this information on page p.13 l. 9-10 of the revised manuscript.

Regarding the difference in reactivation latencies, we agree that somewhat surprisingly, decoding performance peaked earlier in the CC than in the NC condition when using the contrast of 3^rd^ vs 1^st^ repetition in the CC condition. Note, however, that decoding of the target memory in the NC and the CC condition (Figure 2D compared to Figure 2E blue) followed a very similar timecourse overall when comparing against chance. Moreover, there is an earlier peak around 1-1.5 seconds in the NC condition that does not reach significance when using a stringent cluster-based permutation analysis, and the cluster-corrected results are thus not entirely conclusive. Also see our response to point #4 of reviewer #3.

(3) Regarding the theta rhythm in classifier fidelity and its relationship to midline theta-band signal, the current results support the phase separation only at the 3rd repetition. It would be important to see how the rhythm of classifier fidelity changes over repeats as well as its characteristics for the NC condition. Moreover, is there any way to confirm that the theta-band rhythm emerges from the hippocampus?

We agree that these are important points to clarify with further analyses. To address the first point about the dominant frequency of memory reactivation in the various conditions, we used IRASA, following the same procedure as in the manuscript, on the fidelity timecourses in the NC and the CC conditions, and separated them into individual recall repetitions. The resulting spectra of the fidelity values are shown in Figure 3 – supplement 1. Reassuringly, the frequency profile and peak frequency did not show an obvious change across repetitions in either condition. However, the NC condition has a slightly faster frequency profile than the CC, with a peak around 4-5Hz (NC) compared with 3 Hz (CC). On p.15 l. 4-11 of the revised manuscript, we briefly discuss that the faster rhythm in the NC compared to the CC condition could reflect an increase in memory load in the CC condition, with participants keeping two memories in mind, consistent with reports of a corresponding slow-down of the theta frequency in the working memory literature (Wolinski et al., 2018).

To more thoroughly analyse the regional specificity of the hippocampal theta and to validate our source analysis, we conducted an additional analysis at source level where we compare the frequency spectrum of our hippocampal region of interest (ROI) with two control areas (superior occipital lobe and primary motor cortex) that should not show memory-related theta oscillations. Figure 3 – supplement 2C-D shows the difference in the raw power spectrum between the hippocampal ROI and each of the two control regions, averaged across 0-2 seconds after cue onset, and showing power differences at frequencies from 1 to 30Hz for visualisation purposes. To statistically evaluate differences in the theta band, we averaged over 3-8 Hz (the frequency band consistently used throughout the manuscript) and used a one-sided t-test to compare the spectrum against that of each control region (Bonferroni-correcting the p-level for the two repeated comparisons). We find that theta power in the hippocampus is significantly increased compared to superior occipital cortex (t(20) = 2.37, p_corrected_ = .0279), with a maximal power difference at 4 Hz, and a qualitatively similar but non-significant increase when contrasting power over hippocampal and precentral virtual channels (t(20) = 1.98, p_corrected_ = .0622), again with a maximum difference at 4Hz. The new results are described on p. 16 l. 6-9 of the revised manuscript.

We hope these two results will assure the reviewers/readers that (1) our source localization approach is able to isolate distinct frequency profiles in different ROIs, with the expected theta increase in the hippocampus; and (2) that the frequency profile of the classifier is very similar across repetitions and thus unlikely to contaminate the phase-modulation results. Of course, intracranial EEG data would be required to unequivocally pinpoint the exact source(s) of the 2-4Hz theta rhythm.

(4) There are other behavioral analyses that could be done to relate the neural findings to the behavior in addition to the intrusion effect (for example: probability of intrusion, RT for memory recall, overall accuracy, etc.). Please see details below in the reviewers' comments.

We thank the reviewers for pointing out several interesting analyses that provided further understanding of the results. In response to reviewer 2’s point 3, we conducted an analysis relating neural competitor reactivation to the level of behavioural intrusions. We extracted the average decoding accuracy of competitor memories across repetitions from 0 to 1 second after cue onset (following Newman and Norman, (2010) as suggested by the reviewer) and correlated this neural index with the probability of intrusions in the first repetition. A significant positive correlation (r(1,20) = .446, p = .04) was found indicating that stronger neural reactivation of the competitor memory was related to a higher probability of intrusions. See p.13 l. 23 – p. 14 l. 4. This new finding also strengthens the point that classification performance is meaningfully related to behaviour (see Essential Revisions #1 above).

Second, in response to reviewer 3’s point #1, we further investigated potential changes in behavioural accuracy across repetitions in the CC condition, and found no significant change, (Z = -.73, p = .47; Wilcoxon signed-rank test of linear slope against zero). This result is in line with the previously reported result on the intrusion score and is now reported in the revised manuscript on p.8 l.2-5 and as a Figure 2 – supplement 1A.

Lastly, in response to reviewer 3’s point #3, we analysed the timing of the subjective button presses and how it changes across repetitions. Response times significantly shortened in all conditions (NC2: rep1: 2.01 sec, rep2: 1.75 sec, rep3:1.68 sec; Z = -4.17, p <.01; CC: rep1: 2.29 sec, rep2: 2.05 sec, rep3:2.04 sec; Z = -2.89, p = .004; NC1: rep1: 2.55 sec, rep2: 2.37 sec, rep3:2.26 sec; Z = 3.03, p = .003; z-values are based on Wilcoxon signed-rank tests of linear slope against zero). These results are mainly included for descriptive purposes and show that response times roughly coincided with the temporal latency of our decoding timecourses. We report this response time analysis on p.8 l. 10-16 of the revised manuscript.

(5) The authors could consider adding representative data examples to illustrate the phase separation findings.

In response to this comment, and for transparency, we now show in Figure 4 – supplement 1 each participant’s fidelity timecourse of target decoding relative to competitor decoding, filtered at the relevant 3Hz frequency. Visual inspection shows that target and competitor decoding is often phase shifted and does generally not tend to go hand in hand over the average trial timecourse. To quantify this phase shift, we subtracted, for each participant and time bin, the phase angle of the target from the angle of the competitor (in complex space). The resulting phase separation is highly consistent with the one we reported in the original manuscript when relating the peak fidelity values to the hippocampal 3Hz phase.

We also conducted an additional cross-correlation analysis on the fidelity timecourses of target and competitor memories, to further corroborate the phase separation on a single-trial level. This analysis revealed a significant and temporally extended cluster of phase lag starting approximately 1 sec after cue onset and lasting until the end of the trial, suggesting that competitor reactivation followed target reactivation by a lag of on average 30 degrees on the 3Hz theta cycle. These new findings can be found in revised Figure 4C of the main manuscript.

We hope that together, these illustrative changes and complementary analyses will provide further understanding of the phase separation findings. Note that the fidelity timecourses in the original manuscript (previously Figure 1D) served a purely illustrative purpose and showed simulated not real data. We understand that this illustration was more confusing than helpful, and changed this figure panel to more clearly represent the predictions based on the Norman et al., (2006) model of interference resolution.

Reviewer #1 (Recommendations for the authors):1. The decoding performance didn't reach statistical significance for both target and competitor (Figure 2e), which makes me skeptical to what extent the decoding courses truly denote the memory reactivations. If not, how could the phase-based memory reactivation conclusion be supported? I noticed that the effect mainly occurred at the 3rd repetition recall. Thus, I would recommend that the authors should at least provide evidence that the decoding performance was significant for the target and competitor in the 3rd repetition.

We agree with the reviewer and now include additional evidence demonstrating that our decoding algorithm is able to pick up memory reactivations with significant above-chance accuracy, including in the competitive condition. Note that while this analysis was not pre-registered, it will hopefully assure the reviewers/readers that we are looking at meaningful reactivation peaks when relating classifier performance to theta phase. In the new analysis, we realigned the decoding timelines to the time point of subjective recollection (i.e., button press to indicate the memory was recalled), instead of cue onset as for all pre-registered analyses. When doing so (see Author response image 1), we do find significant clusters of target classification accuracy preceding the button press by approx. 200ms. In Author response image 1, we also show that target classification on correct trials is significantly stronger than on incorrect trials, indicating that our method of detecting memory reinstatement is sensitive to behavioural memory success. Lastly, we also find evidence (though not surviving cluster-correction) of competitor reactivation when only using incorrect trials, both response-locked (Author response image 1) and cue-locked (Author response image 1), the former again in a similar time window around 200ms before response.

The most likely reason why a cue-locked decoding approach produces less robust decoding accuracy is the considerable variance in the timing of memory recall (and thus neural pattern reinstatement) across trials, conditions, and participants (see also Reviewer #3 point #3, where we show that RTs differ across repetitions). Such temporal variability is rectified to some degree when aligning the timeline to the button press. Having said that, there are several reasons for why we pre-registered our analyses locked to the onset of the cue. The most important reason is that the phase of theta oscillations is thought to be reset by a retrieval cue (see: Rizzuto et al., 2006; ter Wal et al., 2021), and the locking of memory reactivations to the theta rhythm can thus be expected to be most coherent when aligned to cue onset. Importantly, we conducted the crucial analyses of classifier-to-theta locking on the level of single trial classification peaks, comparing phase shifts of target and competitor peaks. These analyses will thus not be affected by whether or not the reactivation peaks are consistent in time across trials and participants, as they would need to be to reveal robust average cue-locked decoding performance. We added the response-locked target decoding to main Figure 2 (panel g,h) in the revised manuscript, and competitor decoding to Figure 2 – supplement 2D; the description of the new results can be found in the main text on p. 12 l. 15 – 23.

2. The target and competitor showed similar profiles instead of an out-of-phase manner (Figure 1d). Their average decoding performance even showed significant reactivation (supplemental figure 3b). The results seem to be not consistent with the major phase separation findings. Moreover, to flesh out the phase separation results, I would recommend the authors provide typical data examples from their results explicitly illustrating the phase separation profiles, e.g., a representative subject, etc.

We thank the reviewer for pointing out these apparent contradictions. First, it is important to note that Figure 1D served a purely illustrative purpose and shows simulated not real data. In the revised figure, we have now made this panel easier to interpret. Also note that, following the same reasoning as in our response to point #1, a trial-by-trial phase shift between target and competitor reactivation peaks would not necessarily be visible in the cross-subject average of decoding performance, because the timing and absolute phase of target and competitor reactivation can vastly differ in each participant. The average shown in the previous manuscript can only roughly indicate the approximate time window where decoding tends to increase when averaging across all subjects.

To flesh out the phase separation results, we followed the reviewer’s recommendation and now show each individual’s fidelity time course of target and competitor decoding in Figure 4 – supplement. Visual inspection shows that target and competitor reactivation is temporally shifted in most participants, often even showing phase opposition. We also conducted a set of new analyses to quantify this phase shift at the level of single trials and single participants. First, we filtered each participant’s decoding fidelity time course at the dominant 3Hz frequency, separately for target and competitor decoding in the 3^rd^ repetition of the competitive condition. We then subtracted, for each time bin, the phase angle of the target from the angle of the competitor (in complex space), and averaged the phase difference on a group level. The resulting phase separation looks very similar to what we report in the original manuscript when relating the peak fidelity to the hippocampal 3Hz phase. Another way of quantifying the phase lag is to compute a cross-correlation between the two signals, which we did using a sliding-window (to preserve some time information) and calculating the cross-correlation for each 330ms time bin. This analysis revealed a significant cross-correlation cluster starting around 1 sec after cue onset and lasting until the end of the trial, with a maximum lag of 30 degrees (Figure 4C). Together, the two new analyses are highly consistent with the phase shift reported in our pre-registered analysis.

3. The key hypothesis is that the neural representation of the target would become stronger, and phase separation would become more prominent across repetitions. They did find stronger reactivation and phase separation for the 3rd repetition compared to the 1st repetition. Meanwhile, is there any corresponding behavioral evidence such that the overlapping memories become less distracting across repetitions?

This was indeed one of the pre-registered behavioural contrasts and reported on p.8 l.1 of the original manuscript. Contrary to our predictions, we found no significant difference in intrusion scores across repetitions, and this relatively stable pattern of intrusions across repeats might be due to the lack of feedback in our task, as discussed on p.23 l.19-22 (also see Reviewer 3’s comment 2; Figure 5).

4. There is no test to confirm the significant theta rhythm in the hippocampus detected in the present study. I think the authors should provide neural evidence backing up that the theta-band rhythm analyzed in their study indeed derives from the hippocampus.

We agree that the original manuscript was lacking evidence for the specificity of the hippocampal theta. We conducted a new analysis to address this concern, computing the frequency spectrum of the raw (trial) signals extracted from virtual sensors in the hippocampus, and comparing it to the spectrum of two control areas (superior occipital lobe and primary motor cortex) that should not show memory-related theta oscillations. Figure 3 – supplement 2 shows the difference in the power spectrum between the hippocampal ROI and each of the control regions, averaged across 0-2 seconds after cue onset, and showing power differences at frequencies from 1 to 30Hz for visualisation purposes. To statistically evaluate differences in the theta band, we averaged power over 3-8 Hz (the band consistently used throughout the manuscript). We find that theta power in the hippocampus is significantly increased compared to superior occipital cortex (t(20) = 2.37, p_corrected_ = .0279), with a maximal power difference at 4 Hz. A qualitatively similar but non-significant increase was found when contrasting the spectrum from hippocampal and precentral virtual channels (t(20) = 1.98, p_corrected_ = .0622), again with a maximum difference at 4Hz. We hope these results will assure the reviewers/readers that our source localization approach is able to isolate distinct frequency profiles in different ROIs, with the expected theta increase in the hippocampus. We added the new results to the result section (p.16 l.5-8) and added two panels to Figure 3 – supplement 2. Of course, MEG source localisation is still an imperfect reconstruction, and we emphasize that while we can show some degree of specificity, intracranial EEG data would be required to unequivocally locate the main source(s) of the 2-4Hz theta rhythm.

Reviewer #2 (Recommendations for the authors):I found much to appreciate about this manuscript and work. Much of that is summarized in my 'Public review' so I won't repeat it here.I had no 'substantial concerns' about the quality of the work or interpretations presented. That is to say, none of the following thoughts that I'll share next should be seen as show-stopping. I will share these thoughts nonetheless in case they can support the growth of this work.– The reasons why the reactivation effects should only become visible in the 3rd repetition were not clear to me. I struggle to come up with a just-so story based on the Norman et al. 2006 model or otherwise to understand these. Some assistance with this could help.

We appreciate this comment and clarify our reasoning in the revised manuscript (p. 4 l.8-21). First, it should be highlighted that in our pre-registration, we hypothesized (i) an increase in target and a decrease in competitor reactivation (i.e., decodability) over repetitions; and (ii) a target-competitor phase difference that exists throughout the task and would increase across repetitions. These hypotheses were only partly confirmed, and we hope we are sufficiently transparent about this in the manuscript. The first hypothesis was motivated by a wealth of empirical studies on retrieval-induced enhancement of target memories and retrieval-induced forgetting of competitor memories (Anderson et al., 1994; Wimber et al., 2015). It can also be deduced indirectly from the Norman et al., (2006) model, which assumes that over time and repetitions of the presumed theta cycle dynamics, weak target nodes are strengthened while overly strong competitor nodes are weakened, leading to better differentiation hence enhanced decodability of the target memory. We now visualise this reasoning in the revised hypothesis graph in Figure 1D, where targets become decodable at increasingly early (higher inhibition) theta phases with gradually less interference from competitors.

The second hypothesis of increasing phase separation over time is based directly on the model’s dynamics. Weak target nodes do not survive high inhibition initially and thus only activate during medium and lower inhibition phases of the theta cycle, however, with repeated strengthening they will become active at an increasingly early, higher-inhibition phase. Therefore, while early in time the target and competitor memories overlap in their peak reactivation phase, this overlap gets reduced by the strengthening and weakening dynamics in the model. Critically, Norman et al. (2006) model these dynamics across several repetitions of a theta cycle, while we instead opted to measure them across recall repetitions, simply because we were not confident that our analysis tools could provide a stable phase estimate for target and competitor reactivation on each theta cycle within a retrieval trial.

– There is a strange discrepancy in the frequency band of the frontal midline theta (~7 Hz) and the frequency of the phase modulated reactivations. I did not see any attempt to reconcile this. Are they simply two totally different things, the 7Hz, and 3Hz thetas?

This is a valid observation and indeed something we did not discuss in the original manuscript. To our knowledge, the relationship between the conflict-related frontal midline theta and the memory-related 2-4Hz theta is still unknown, though both rhythms are consistently found in their respective cognitive domains. The frontal midline theta is typically found at 7-8Hz, like in our own study, and has been related to cognitive conflict/control across domains (Cavanagh and Frank, 2014). In the memory domain, frontal midline theta amplitudes increase with increased competition and decrease with downregulation of the competitor memory (Ferreira et al., 2014; Hanslmayr et al., 2010). Our own finding that power in this frontal theta rhythm increases under conditions of high competition is thus a replication of previous work, and a sanity check that physiological markers of competition increase in the CC compared to the NC condition. A memory-related theta in the human hippocampus is frequently found around 2-3Hz in intracranial recordings (Goyal et al., 2020; Lega et al., 2012; ter Wal et al., 2021), and thus slower than the dominant theta rhythm in the rodent hippocampus. Our findings suggest that the phase separation of competing memories relates to this slower hippocampal rhythm, not the faster frontal rhythm. Beyond this observation, we are not aware of literature linking the two theta rhythms directly, though this would certainly be an interesting field for future studies. Hence, for the time being, we treat these two rhythms as separate phenomena, and clarify our reasoning on p.9 l.11-14 of the revised manuscript.

– There are other tests that could be done to relate the reactivation dynamics to the behavior than are described. For example, one could compare the degree of reactivation to the probability of intrusion. Newman et al., 2010 – https://pubmed.ncbi.nlm.nih.gov/20181622/ – for example did something similar. At the coarsest level, the probability of an intrusion should be lowest for trials where there was no evidence of competitor reactivation.

Thank you for this suggestion. Taking inspiration from the suggested method, we extracted the average decoding across repetitions for competitor memories (using the same trials as in the manuscript) from 0 to 1 second after cue onset (following Newman and Norman, (2010)) and correlated this with the probability of intrusions in the first repetition. This yielded a significant correlation (r(1,20) = .446, p = .04), such that stronger reactivation of the competitor memory was associated with a higher probability of intrusions. This new finding also adds evidence in support of the decoding performance being related to behaviour. See Figure 2 – supplement 1C and p.13 l. 23 – p.14 l.4, of the revised manuscript.

Reviewer #3 (Recommendations for the authors):There are some suggestions that could improve the overall conclusions one can draw from the manuscript.The manipulation of repeating the retrieval period is interesting and allows for some novel hypotheses and questions. The authors use a measure of intrusions (by assessing how often subjects select the competitor subordinate category) and find that this is independent of the recall trial number. How about accuracy for the CC conditions?

We thank the reviewer for this suggestion. We did not find a change in behavioural accuracy of CC across repetitions, (Z = -.73, p = .47; Wilcoxon signed-rank test of linear slope against zero). This additional data point is now reported in the revised manuscript on p.8 l.2-5 and as a Figure 2 – supplement 1A. Note that while intrusion scores remain at the same level over recall repetitions, we report new analyses showing that the intrusions do relate to behaviour in a sensible way, such that more neural competitor activation is associated with a higher likelihood of intrusions, see response to Reviewer #2’s point 3.

The authors find an increase in fontal theta that lateralizes to the right for the competitive condition (CC). While the authors find the lateralization surprising, and instead expected a greater increase over midline structures, the lateralization here may be more consistent with recent literature implicating the right DLPFC in action inhibition. In this case, the inhibited action may be related to the competitor's memory. Alternatively, if there is a conflict signal that is relevant for this task, although the authors look for this conflict signal during retrieval it may also be helpful to identify whether a conflict signal is present during the encoding portion of the task when the competitor memory is introduced.

We thank the reviewer for these comments and address them as follows. Regarding the right lateralization of the ‘conflict theta’, we added a section to the main results (p. 9 l. 9-11) to highlight that response conflict and response inhibition are indeed often found to engage a right-lateralised frontal network. Regarding the conflict signal during the encoding phase, we agree that a thorough analysis of the memory (re)activation processes occurring already during encoding is very interesting and we are planning such analyses in the future. We do think that a full analysis of the encoding data is beyond the scope of the current paper. Having said that, in response to this reviewer comment, we analysed theta power when participants encode a second competing memory (CC condition) in the learning phase, compared to encoding the same associate a second time. This analysis indeed reveals a non-significant theta power increase in the CC condition with a maximum around 100-300ms after second picture onset, which could be interpreted as conflict signal (see Author response image 2). Note, however, that during encoding, this contrast is naturally confounded with repetition (NC2) vs no-repetition (CC) of an image, and can therefore not isolate the pure competition component. We decided to not include these analyses in the revised manuscript for the time being, although we would be happy to include them in the supplements if the reviewers/editors consider them central to the conclusions.

**Author response image 2. sa2fig2:** Conflict theta during second presentation at encoding. We found that there is stronger theta power when participants encode a second competing associate (target item) with an old memory cue in the CC condition, compared to re-encoding the first associate with the old memory cue in the NC condition. X-axis shows time from image onset (0) and one second after the image appeared on screen.

One of the advantages of the task design is that the subordinate categories allow for classifiers to be built that can decode which memory is being reactivated. The authors use an LDA-based classifier on the MEG sensor amplitudes to construct and test the classifiers. Interestingly, when decoding the retrieval data, the classifiers are significant in the NC conditions a full 2.5 seconds after the cue. This seems like a very long time, as the authors acknowledge. In the description of the task, the authors report that the subjects indicate when they have the picture in mind, and then indicate the supra- and subordinate categories. Presumably, then, this means there is a response time. How does this activation compare to the response time?

We thank the reviewer for highlighting this observation, and we further investigated the temporal relationship between memory reactivation and response times in a number of new analyses. Response times for the subjective recollection button press were on average 2.11 sec, and the peak decodability therefore indeed coincides roughly with the response. Response times significantly shortened in all conditions across the three repetitions (NC2: rep1: 2.01 sec, rep2: 1.75 sec, rep3:1.68 sec; Z = -4.17, p <.01; CC: rep1: 2.29 sec, rep2: 2.05 sec, rep3:2.04 sec; Z = -2.89, p = .004; NC1: rep1: 2.55 sec, rep2: 2.37 sec, rep3:2.26 sec; Z = 3.03, p = .003; Wilcoxon signed-rank tests of linear slope against zero). In response to this reviewer comment and Reviewer #1’s point #1 (see Author response image 2 and the above response), we performed an additional analysis locking the decoding to response onset rather than cue onset. This analysis reveals significant clusters of memory reactivation (i.e. decoding peaks) approximately 200ms before the subjective recollection response. We report these new analyses on p.8 l. 10-16 of the revised manuscript.

The authors report that there is no significant reinstatement of the target category during correct competitive trials as compared to the competitor category, although they claim that the classifier performance peaks at a similar time to the NC condition. This seems a bit concerning because one would expect to see some evidence of reactivation if subjects are making the correct decision. How should this be reconciled? Instead, the authors find that the evidence for target memory increases over recalls, which they offer as evidence that these memories are increasingly being reactivated. However, classifier performance peaks at a different time than when you see reinstatement in NC condition and is in fact earlier. Why?

We appreciate that the evidence for target reactivation in the CC was not sufficiently strong based on the (pre-registered) analyses reported previously. As the reviewer also points out, decoding performance peaked earlier in the CC than in the NC condition, at least when based on the comparison of 3^rd^ vs 1^st^ repetition (where on the 3^rd^ repetition, memory recall might indeed have happened earlier than on the 1^st^ repetition). When comparing target decoding against chance level in the NC and CC conditions (Figure 2D and 2E), they do follow very similar timecourses. Moreover, an earlier peak around 1-1.5 seconds can also be seen in the NC condition but does not reach significance when using a stringent cluster-based permutation statistic, and the results are thus not entirely conclusive. Addressing the concern about general decoding performance and relationship to behaviour, we conducted the above-mentioned response-locked analyses, showing a clear reactivation peak just before the response when aligning trials to the time point of subjective recollection. Moreover, and relevant to the specific point raised here, we also show that this target reactivation is significantly stronger when participants make a correct vs incorrect decision, suggesting there is more neural evidence for the target class when participants correctly retrieve this class. More generally, response-locked reactivation appears to be more robust suggesting to us that there is considerable variance in the timing of memory recall across trials, conditions, and participants (also see Essential Revisions point #1, Reviewer #1 point #1, and Reviewer #3 point #3), and such variability makes it difficult to see clear cue-locked reactivation peaks on. Corresponding changes in the manuscript occur in the revised Figure 2 and in the main text on p. 12 l. (15 – 23).

It would be helpful if the authors could please clarify Supp Figure 3B. What does it mean when they collapse targets and competitors? Do they mean that the classifier can decode up either one?

We understand that this point needs clarification. Of course, the design of our study and all our main analyses are tailored to separate target and competitor reactivation. This specific analysis collapsing across target and competitor reactivation was conducted purely for the purpose of identifying a time window where on average, across all trials and participants, target and competitor reactivation tend to be high, such that we could then use this unbiased time window (i.e., not biased towards target or competitor decoding) to probe for the hypothesized up and down-regulation of target and competitor memories, respectively, across repetitions (as seen in Figure 2 – supplement 2B and Figure 2F). For all other analyses, we did keep target and competitor decoding separate. We clarify this reasoning now in a sentence on p.49 l. 8-12. See also Essential Revisions point #1.

If the authors are seeing increasing evidence of target information over several repeats, then this raises the question as to whether the classifier of subordinate target category would only work if only looking at the third repeat? The authors test this by comparing classifier performance in the 1st v 3rd repeat. However, how about comparing performance in the 3rd repeat versus chance?

We followed the reviewer’s suggestion, and we do also find significant decoding performance for target memories when comparing against chance (50%) in the third repetition, with a significant cluster emerging from 1.81 to 2.08 seconds after cue onset. However, this cluster does not survive a more stringent cluster-based permutation correction for multiple comparisons across time (p_corr_ = .07). We added this analysis on page p.13 l. 9-10. See also Essential Revisions point #2.

The authors then tie in classifier decoding with the phase of the theta rhythm in the hippocampus. First, the classifier fidelity itself appears to have a 3 Hz rhythm. They then compute the modulation index (MI) of classifier fidelity to the hippocampal 3Hz phase. Classifier fidelity is only modulated by the hippocampal phase during the 3rd retrieval repetition. Similarly, we only see significant phase differences between target and competitor in the last repetition. This would seem to support the hypothesis that repetition leads to greater separation and better memory. However, it would be then helpful to know how the rhythm of classifier fidelity (the underlying 3Hz rhythm of classifier performance) changes across repeats. Is this fixed, or does this also exhibit changes with repetition? Is this also the case in the NC condition? For the NC condition, how do the phases modulation compare with the phases of target and competitors in the CC condition?

We agree that changes in classifier-to-phase locking over time might be ‘contaminated’ by frequency changes. We thus followed the reviewer’s suggestion to analyse the dominant frequency in the fidelity timecourses in both NC and CC across repetitions. Both conditions show peaks in the theta range, though the NC condition has a slightly faster frequency profile than the CC: 4-5Hz (NC) compared with 3 Hz (CC). Reassuringly, there is no obvious change in the peak frequency across repetitions in either condition. We now show these spectra of the fidelity values in Figure 3 – supplement 1. Note that there is a potentially interesting explanation for the faster rhythm in the NC than the CC condition, such that the increased memory load in the CC (with participants keeping two memories in mind) being adapted to by a slow-down of the theta frequency as previously reported in the working memory literature (Wolinski et al., 2018). We briefly allude to this possibility on p.15 l. 4-11 of the revised result section and p.28 l.23 – p.29 l.4 of the discussion. See also Essential Revisions #3.

References

Anderson, M. C., Bjork, R. A., and Bjork, E. L. (1994). Remembering can cause forgetting: Retrieval dynamics in long-term memory. *Journal of Experimental Psychology. Learning, Memory, and Cognition*, *20*(5), 1063–1087. https://doi.org/10.1037//0278-7393.20.5.1063

Cavanagh, J. F., and Frank, M. J. (2014). Frontal theta as a mechanism for cognitive control. *Trends in Cognitive Sciences*, *18*(8), 414–421. https://doi.org/10.1016/j.tics.2014.04.012

Cohen, M. X. (2014). *Analyzing Neural Time Series Data: Theory and Practice*. MIT Press.

Ferreira, C. S., Marful, A., Staudigl, T., Bajo, T., and Hanslmayr, S. (2014). Medial prefrontal theta oscillations track the time course of interference during selective memory retrieval. *Journal of Cognitive Neuroscience*, *26*(4), 777–791. https://doi.org/10.1162/jocn_a_00523

Goyal, A., Miller, J., Qasim, S. E., Watrous, A. J., Zhang, H., Stein, J. M., Inman, C. S., Gross, R. E., Willie, J. T., Lega, B., Lin, J.-J., Sharan, A., Wu, C., Sperling, M. R., Sheth, S. A., McKhann, G. M., Smith, E. H., Schevon, C., and Jacobs, J. (2020). Functionally distinct high and low theta oscillations in the human hippocampus. *Nature Communications*, *11*(1), 2469. https://doi.org/10.1038/s41467-020-15670-6

Hanslmayr, S., Staudigl, T., Aslan, A., and Bäuml, K.-H. (2010). Theta oscillations predict the detrimental effects of memory retrieval. *Cognitive, Affective and Behavioral Neuroscience*, *10*(3), 329–338. https://doi.org/10.3758/CABN.10.3.329

Kerrén, C., Linde-Domingo, J., Hanslmayr, S., and Wimber, M. (2018). An Optimal Oscillatory Phase for Pattern Reactivation during Memory Retrieval. *Current Biology: CB*, *28*(21), 3383-3392.e6. https://doi.org/10.1016/j.cub.2018.08.065

Lega, B. C., Jacobs, J., and Kahana, M. (2012). Human hippocampal theta oscillations and the formation of episodic memories. *Hippocampus*, *22*(4), 748–761. https://doi.org/10.1002/hipo.20937

Linde-Domingo, J., Treder, M. S., Kerrén, C., and Wimber, M. (2019). Evidence that neural information flow is reversed between object perception and object reconstruction from memory. *Nature Communications*, *10*(1), 179. https://doi.org/10.1038/s41467-018-08080-2

Newman, E. L., and Norman, K. A. (2010). Moderate Excitation Leads to Weakening of Perceptual Representations. *Cerebral Cortex (New York, NY)*, *20*(11), 2760–2770. https://doi.org/10.1093/cercor/bhq021

Norman, K. A., Newman, E., Detre, G., and Polyn, S. (2006). How Inhibitory Oscillations Can Train Neural Networks and Punish Competitors. *Neural Computation*, *18*(7), 1577–1610. https://doi.org/10.1162/neco.2006.18.7.1577

Rizzuto, D. S., Madsen, J. R., Bromfield, E. B., Schulze-Bonhage, A., and Kahana, M. J. (2006). Human neocortical oscillations exhibit theta phase differences between encoding and retrieval. *NeuroImage*, *31*(3), 1352–1358. https://doi.org/10.1016/j.neuroimage.2006.01.009

Staresina, B. P., and Wimber, M. (2019). A Neural Chronometry of Memory Recall. *Trends in Cognitive Sciences*, *23*(12), 1071–1085. https://doi.org/10.1016/j.tics.2019.09.011

ter Wal, M., Linde-Domingo, J., Lifanov, J., Roux, F., Kolibius, L. D., Gollwitzer, S., Lang, J., Hamer, H., Rollings, D., Sawlani, V., Chelvarajah, R., Staresina, B., Hanslmayr, S., and Wimber, M. (2021). Theta rhythmicity governs human behavior and hippocampal signals during memory-dependent tasks. *Nature Communications*, *12*(1), 7048. https://doi.org/10.1038/s41467-021-27323-3

Wimber, M., Alink, A., Charest, I., Kriegeskorte, N., and Anderson, M. C. (2015). Retrieval induces adaptive forgetting of competing memories via cortical pattern suppression. *Nature Neuroscience*, *18*(4), 582–589. https://doi.org/10.1038/nn.3973

Wolinski, N., Cooper, N. R., Sauseng, P., and Romei, V. (2018). The speed of parietal theta frequency drives visuospatial working memory capacity. *PLOS Biology*, *16*(3), e2005348. https://doi.org/10.1371/journal.pbio.2005348